# Folliculin regulates mTORC1/2 and WNT pathways in early human pluripotency

J. Mathieu[1,2,3], D. Detraux[1,2,8], D. Kuppers[4], Y. Wang[2,5], C. Cavanaugh[2,3], S. Sidhu[1,2], S. Levy[1,2], A.M. Robitaille[2,6], A. Ferreccio[1,2], T. Bottorff[1,2], A. McAlister[1,2], L. Somasundaram[1,2], F. Artoni[1,2], S. Battle[2,7], R.D. Hawkins[2,7], R.T. Moon[2,6], C.B. Ware[2,3], P.J. Paddison[2,4] & H. Ruohola-Baker [1,2]

To reveal how cells exit human pluripotency, we designed a CRISPR-Cas9 screen exploiting the metabolic and epigenetic differences between naïve and primed pluripotent cells. We identify the tumor suppressor, Folliculin(FLCN) as a critical gene required for the exit from human pluripotency. Here we show that *FLCN* Knock-out (KO) hESCs maintain the naïve pluripotent state but cannot exit the state since the critical transcription factor TFE3 remains active in the nucleus. TFE3 targets up-regulated in *FLCN* KO exit assay are members of Wnt pathway and ESRRB. Treatment of *FLCN* KO hESC with a Wnt inhibitor, but not *ESRRB/FLCN* double mutant, rescues the cells, allowing the exit from the naïve state. Using co-immunoprecipitation and mass spectrometry analysis we identify unique FLCN binding partners. The interactions of FLCN with components of the mTOR pathway (mTORC1 and mTORC2) reveal a mechanism of FLCN function during exit from naïve pluripotency.

[1] Department of Biochemistry, University of Washington, Seattle, WA 98195, USA. [2] Institute for Stem Cell and Regenerative Medicine, University of Washington, Seattle, WA 98109, USA. [3] Department of Comparative Medicine, University of Washington, Seattle, WA 98109, USA. [4] Human Biology Division, Fred Hutchinson Cancer Research Center, Seattle, WA 98109, USA. [5] Paul G. Allen School of Computer Science & Engineering, University of Washington, Seattle, WA 98109, USA. [6] Department of Pharmacology, University of Washington, Seattle, WA 98195, USA. [7] Department of Medical Genetics & Genome Sciences, University of Washington, Seattle, WA 98195, USA. [8] Present address: Laboratory of Cellular Biochemistry and Biology (URBC), University of Namur, Namur 5000, Belgium. Correspondence and requests for materials should be addressed to P.J.P. (email: paddison@fredhutch.org) or to H.R.-B. (email: hannele@uw.edu)

Unveiling the molecular mechanisms through which pluripotency is maintained holds promise for understanding early animal development, as well as developing regenerative medicine and cellular therapies. Pluripotency does not represent a single defined stage in vivo. Following implantation, pluripotent naïve epiblast cells transition to a pluripotent stage primed toward lineage specification. Those subtle stages of pluripotency, with similarities and differences in measurable characteristics relating to gene expression and cellular phenotype, provide an experimental system for studying potential key regulators that constrain or expand the developmental capacity of ESC[1–12]. While multiple pluripotent states have been stabilized from early mouse and human embryos, it is not fully understood what regulates the transitions between these states.

The molecular mechanisms and signaling pathways involved in the maintenance and exit from naïve pluripotency have been extensively studied in mouse, but are still poorly understood in human[13]. In mouse, the naive pluripotency program is controlled by a complex network of transcription factors, including Oct4, Sox2, Nanog, Klf2/4/5, Tfcp2l1 (Lbp9), Prdm14, Foxd3, Tbx3, and Esrrb[14–18]. Interestingly, Esrrb has been shown to regulate the naïve pluripotent state in mouse[19,20], but RNAseq data suggest that existing human ESC lines lack robust expression of Esrrb[1,6,7,11,12,21].

Naïve and primed pluripotent cells have important metabolic and epigenetic differences[1,12,22,23,24]. We utilize these differences to design a functional CRISPR-Cas9 screen to identify genes that promote the exit from human naïve pluripotency. In the screen, we identify folliculin (FLCN) as one of the genes regulating the exit. FLCN knockout naïve hESC remain pluripotent since they retain high levels of the pluripotency marker, OCT4, and early naïve markers (KLF4, TFCP2L1, DNMT3L). However, we show a requirement for FLCN to exit the naïve state. During normal exit from naïve pluripotency, the transcription factor TFE3 is excluded from the nucleus, while in FLCN KO hESC TFE3 remains nuclear, maintaining activation of naïve pluripotency targets. ESRRB KO in FLCN KO hESC does not rescue the phenotypes. However, we find that TFE3 targets involved in Wnt pathway are up-regulated in FLCN KO and inhibition of Wnt restores the exit from the naïve state in FLCN KO cells. Mass spectrometry analysis reveals that FLCN binds to different proteins in the naïve state and upon exit from the naïve state, allowing us to propose a new model for the action of FLCN in early pluripotent states.

## Results

**CRISPR KO screen during exit from human naïve pluripotency**. S-adenosyl methionine (SAM) levels, controlled by nicotinamide N-methyltransferase (NNMT), are critical during the naïve-to-primed hESC transition, where the epigenetic landscape changes through increased H3K27me3 repressive marks[1,24]. Using CRISPR-Cas9 technology, we generated NNMT KO naïve hESC lines[1]. As expected, SAM levels and H3K27me3 marks are increased in NNMT KO naïve cells compared to wild type naïve cells[1] (Fig. 1a). Principal component analysis of NNMT KO cells revealed that their gene expression signature shifts towards the primed stage, even when grown in naïve-like culture conditions (2iL-I-F)[1] (Supplementary Fig. 1A). However, NNMT KO cells exhibit only a partial primed gene expression signature. We found that 913 genes in NNMT KO do not display the expected methylation pattern for primed cells (Fig. 1b; Supplementary Data 1A) and 1967 genes down-regulated in primed hESC fail to decrease expression in NNMT KO cells (Supplementary Fig. 1B, Supplementary Data 1B). To uncover which factors are regulating these 1967 genes, we tested their potential enrichment as target genes of 45 transcription factors based on ENCODE ChIP-seq

data in primed hESCs (Supplementary Fig. 1C). The most enriched candidate regulators include CTBP2, TAF1, EGR1, TEAD4 (ref. [5]), JUND, SP1, and TFE3, suggesting that these transcription factors are normally repressed during the exit from naïve hESC state.

We performed a CRISPR KO screen to test the functionality of the identified transcriptional regulators, and to identify additional genes required for the exit from the human naïve pluripotent state. To control for hESC cell type variability, we performed the screen with one naïve cell type and validated the key candidates in another naïve line, from a different genetic background. Naïve 2iL-I-F hESC were infected with a lentiviral pool containing a human CRISPR-Cas9 knockout (GeCKO) library targeting 18,080 genes with 64,751 unique sgRNAs[25]. Cells were allowed to exit the naive hESC state, and the primed hESC were selectively eliminated with a mixture of methotrexate and acetaldehyde, exploiting the differences in the metabolic requirements of these cell types[1] (Fig. 1c). Methotrexate inhibits both methionine synthase (MS) and S-adenosyltransferase (MAT) activities, reducing SAM levels[26,27]. Acetaldehyde also inhibits methionine synthase[28]. Primed hESC have higher levels of repressive histone methylation (H3K27me3) marks, increasing their dependence on SAM levels[1,24] and increasing their sensitivity to methotrexate and acetaldehyde treatment (Fig. 1d). Methotrexate/acetaldehyde treatment specifically kills the primed cells and cells transitioning to the primed state (Fig. 1e). To identify the genes required for the primed state transition, we analyzed the surviving cells for enriched sgRNAs (Fig. 1c). We hypothesized that the survivors had inactivation of genes required for exit from naïve pluripotency.

The CRISPR screen successfully identified genes known to regulate human primed pluripotent stem cells (Supplementary Data 2 (ref. [29])), and many novel regulators, including GREB1, FAM60A, UROS, GPI, MED12, TSC2, RELT, CCDC159, NARFL, and FLCN. These potential regulators of the exit from the naïve pluripotent state are involved in metabolic switching, signaling, and chromatin remodeling (Fig. 1f; Supplementary Data 2). As expected for a negative selection screen relying on apoptotic activity, the screen also identified genes required for apoptosis (Supplementary Data 2). Gene hits of interest were individually validated (Fig. 1g; Supplementary Data 2). We opted to further study the role of FLCN, a tumor suppressor mutated in the Birt–Hogg–Dubé syndrome, in early human pluripotent states[30]. Flcn−/− mice are embryonic lethal and exhibit a disorganized epiblast soon after implantation[31]. Flcn acts in mouse ESC by regulating Esrrb function[19]. However, ESRRB expression is low in hESC compared to mESC and it is not yet known whether ESRRB is required in human naïve hESC[21], thus the mode of action for FLCN is not yet understood in human pluripotency.

**FLCN is not required to maintain the naïve pluripotent state**. We investigated the role of FLCN in human pluripotency by generating FLCN KO lines in naïve human ESC. To control for potential differences in published naïve hESC states, we tested FLCN function in different naïve growth conditions and in different cellular backgrounds (growth conditions: 5iLA and 2iL-I-F; cellular background: from the Boston-derived lines we chose WIBR3 and from Seattle-derived Elm/Elf lines (Supplementary Fig. 1D) we chose Elf1; refs. [1,4,32]).

We generated FLCN KO lines in two cellular backgrounds (WIBR3 and Elf1) hESC using four guide RNAs targeting FLCN (Fig. 2a, c; Supplementary Fig. 2A). Guides located in exons 3 and 4 created InDel mutations, introducing STOP codons and resulting in the absence of FLCN protein (Fig. 2a, c; Supplementary Fig. 2A). We then generated a FLCN rescue line by

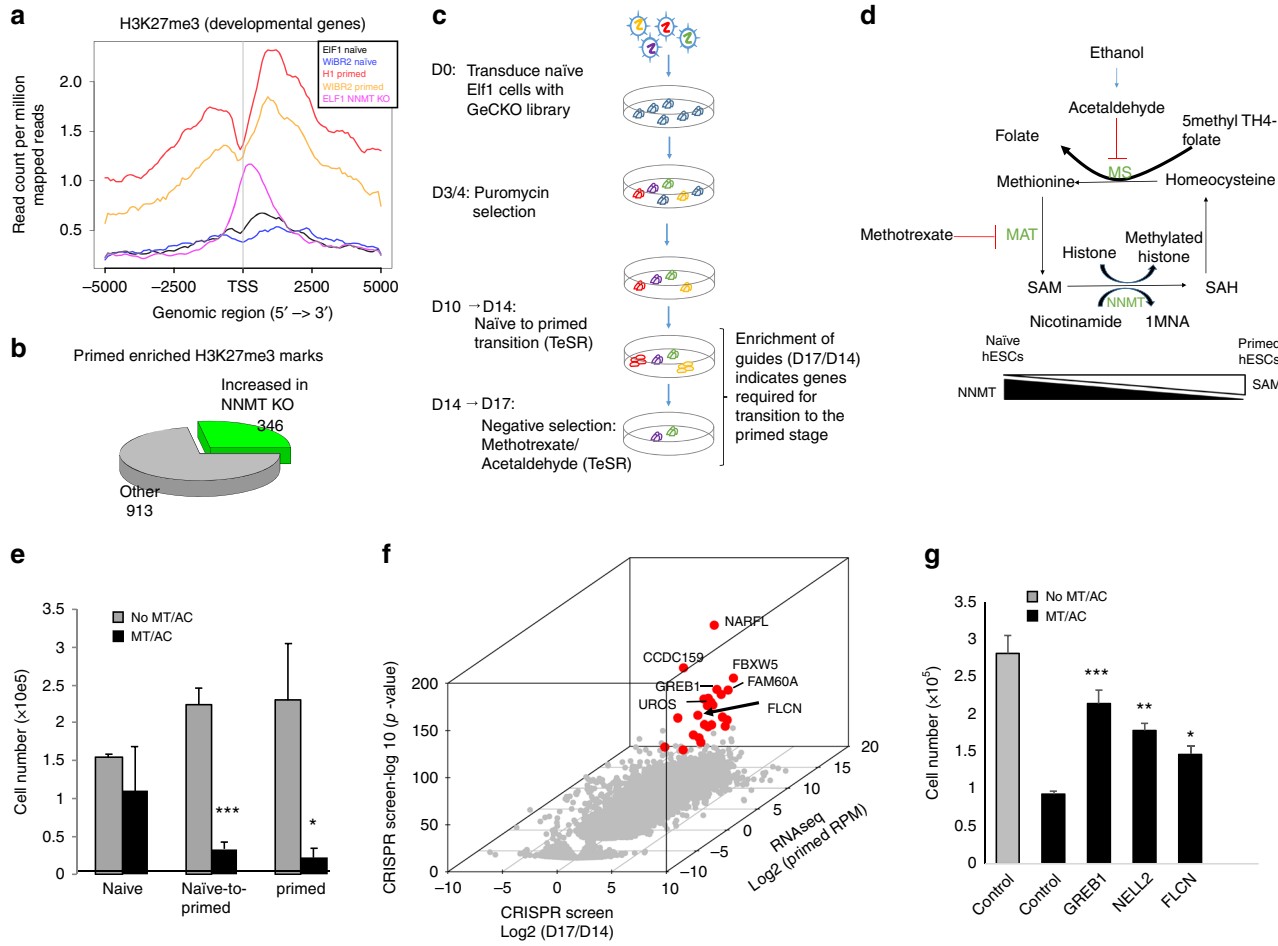

**Fig. 1** CRISPR-Cas9 screen reveals FLCN as an essential gene during the exit from the naïve pluripotent state. **a** H3K27me3 profile 5KB around transcription start site in developmental genes in naïve, primed and naive *NNMT* CRISPR KO samples were plotted from ChIPseq analysis. **b** Only 27% of the genes with H3K27me3 marks enriched in primed vs. naïve hESC are increased in *NNMT* KO naïve hESC. **c** Flowchart of genome wide CRISPR screen for exit of naïve state using GeCKO v2 library. **d** Schematic model of action of methotrexate/acetaldehyde selection. **e** Methotrexate/acetaldehyde selectively kills primed hESC and cells exiting naïve pluripotent state (4D TeSR). S.e.m.; *$p < 0.05$, ***$p < 0.001$; two-tailed *t*-test, $n = 3$ or 4 biological replicates. **f** CRISPR candidate hits selected by integrating CRISPR screen and RNA-seq data in primed hESC[68]. Genes that are significant CRISPR hits (FDR < 0.05, at least 2.5-fold higher in day 17 compared to day 14) and expressed in primed hESC (Elf1 AF, >10 normalized read counts) are colored in red. Apoptotic genes were removed (Supplementary Data 2). **g** Secondary screen. Naïve hESC were infected with lentiCRISPR virus targeting single genes and induced to exit naïve state in TeSR media. The number of surviving cells after 3 days of methotrexate/acethaldehyde (MT/AC) was counted. S.e.m.; *$p < 0.05$, **$p < 0.005$; two-tailed *t*-test, $n = 3$ biological replicates

introducing an inducible FLCN-GFP fusion construct into the AAVS1 safe harbor site of the naïve *FLCN* KO line (Fig. 2b). Overexpression of the FLCN-GFP after doxycycline treatment was shown by western blot analysis (Fig. 2c; Supplementary Fig. 2B). Immunostaining analysis revealed that *FLCN* KO naïve hESC (2iL-I-F and 5iLA) maintain the expression of the pluripotency factor OCT4 (Fig. 2d). We further performed RNAseq analysis to compare gene expression changes in the *FLCN* KO hESC line vs. the rescue hESC line (*FLCN* KO vs. *FLCN* KO + FLCN-GFP; Supplementary Fig. 2C, D, Supplementary Data 3A). In order to assess whether *FLCN* KO hESC are still in the naïve state and to identify potential stage-specific markers among the differentially expressed genes, we utilized RNAseq data from non-human primate, cynomolgus monkey, pre-and post-implantation in vivo blastocysts (*Macaca fascicularis*[33]). We compared the cynomolgus monkey in vivo data to the *FLCN* KO with and without the rescue construct (Fig. 2e; Supplementary Data 3B). Important early naïve pluripotency markers, such as TFCP2L1 (LBP9), DNMT3L, KLF4, DPPA3, and DPPA2 (ref. [13]) are among the 40 genes that are up-regulated both in *FLCN* KO

hESC cell culture and in vivo pre-implantation cynomolgus monkey blastocysts (Fig. 2e, f; Supplementary Fig. 1A, Supplementary Data 3B). t-SNE analysis of the Macaca data confirmed that DNMT3L and TFCP2L1 are highly expressed in pre-implantation stages compared to post-implantation stages (Fig. 2f). We validated the up-regulation of DNMT3L and TFCP2L1 by reverse transcription quantitative PCR (RT-qPCR) in naïve 2iL-I-F and 5iLA *FLCN* KO hESC compared to the rescued lines (Fig. 2g, h). Tfcp2l1 is highly expressed in mouse pre-implantation ICM compared to mouse post-implantation epiblast[34,35] and was proposed as a critical regulator to stabilize both the mouse and the human ESC naïve state[12,36–38]. TFCP2L1 controls transposable elements (TRE)[36], and TRE expression has been proposed as a molecular criterion of early naïve pluripotent stem cells[4]. We analyzed the TRE signature and found that *FLCN* KO naïve (2iL-I-F) hESC expressed a subset of the early naïve (5iLA) TRE (Supplementary Fig. 2E; Supplementary Data 3C).

Our data show that when in the naïve state, *FLCN* KO cells express early naïve markers also seen in monkey pre-implantation pluripotent cells (Fig. 2e–h; Supplementary Fig. 2C, E; e.g.

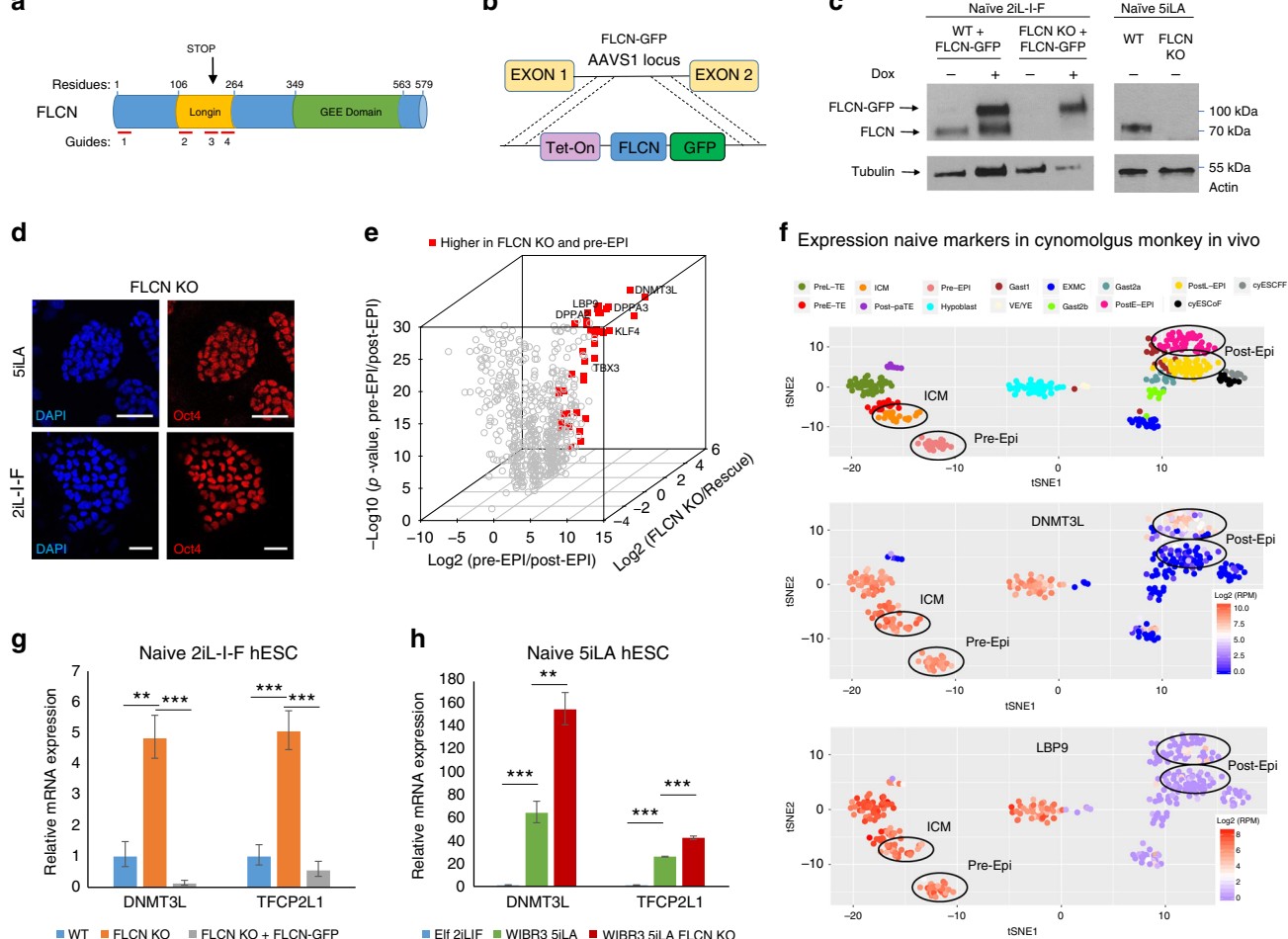

**Fig. 2** *FLCN* KO hESC express naïve pluripotency markers. **a** Schematic representation of FLCN protein and location of CRISPR guide RNAs used. Sanger sequencing analysis of Elf1 2iL-I-F *FLCN* KO hESC reveals introduction of STOP codons in exons 3. **b, c** Generation of WIBR3 5iLA FLCN KO and Elf1 2iL-I-F FLCN KO line, and overexpression of FLCN-GFP fusion protein in wild type and FLCN KO background. **d** The pluripotency marker Oct4 is expressed in FLCN KO 2iL-I-F and 5iLA hESC. Scale bars represent 50 μm. **e** Expression pattern of genes up-regulated by FLCN KO compared to genes up-regulated in monkey pre- vs. post-implantation stage. Red: genes up-regulated in FLCN KO and pre-implantation, known ground state pluripotency markers are labeled. **f** tSNE analysis of naïve markers DNMT3L, TFCP2L1, and ESRRB in in vivo blastocysts from non-human primate, cynomolgus monkey (*Macaca fascicularis*[33]). **g, h** RT-qPCR analysis of naïve markers DNMT3L and TFCP2L1 in naïve 2iL-I-F wild type (WT), FLCN KO, and rescue line (FLCN KO + OE FLCN-GFP) (**g**) and in naïve 5iLA WT and FLCN KO (**h**). S.e.m.; **p < 0.005, ***p < 0.001; two-tailed *t*-test, n = 3–10 biological replicates

DNMT3L, TFCP2L1, and TRE), suggesting that *FLCN* KO upregulates markers of early primate development. Taken together, these results show that FLCN is not required to maintain the naïve pluripotency state.

**FLCN is required for hESC to exit from the naïve state**. To further dissect the function of FLCN in hESC, we used the two naïve hESC conditions (2iL-I-F and 5iLA) and two genetic backgrounds (Elf1 and WIBR3) to analyze exit from the naïve hESC stage (Fig. 3a). We first tested the differential gene expression signatures between the *FLCN* KO mutant and control lines using RNAseq and bioinformatics platforms after naïve hESC(2iL-I-F) had been treated 7 days in TeSR medium to allow exit from naive pluripotency (Supplementary Data 4A). Principal component analysis revealed that while, after 7 days of culture in TeSR, naïve hESC moved toward the primed state, the *FLCN* KO line was not able to fully transition, remaining closer to the naive profile (Fig. 3b). We also compared the 7-day TeSR-treated hESC to the cynomolgus monkey in vivo data[33] and found that genes up-regulated in 7D TESR in *FLCN* KO

hESC are significantly enriched for monkey pre-implantation marker genes (hypergeometric test *p*-value $1.27 \times 10^{-41}$) while genes up-regulated in 7-day TeSR in wild type are significantly enriched for monkey post-implantation marker genes (hypergeometric test *p*-value $5.54 \times 10^{-49}$) (Fig. 3c; Supplementary Data 4B, C). We validated these observations by RT-qPCR and western blot (Fig. 3d–g). When induced to exit the naïve state by omitting the signaling pathway inhibitors and activating FGF pathway using TeSR medium, the *FLCN* KO line failed to exit (naïve marker DNMT3L still expressed, Fig. 3d; primed markers IDO1, LDHA, and HIF1α not upregulated, Fig. 3e, f; Supplementary Fig. 3A; ref. [1]). Overexpression of FLCN rescued these *FLCN* KO phenotypes (Fig. 3d–f). Collectively, these data indicate that FLCN is required for hESC to exit the naïve state.

Interestingly, when *FLCN* KO cells were pushed to exit the naïve state, they did show some positive markers indicative of a partially successful transition: NNMT expression was down-regulated, and JARID2 expression and H3K27me3 marks were upregulated, reflective of primed stage (Fig. 3f–g; see ref. [1]). We tested if overexpression of NNMT (NNMT OE) in the *FLCN* KO

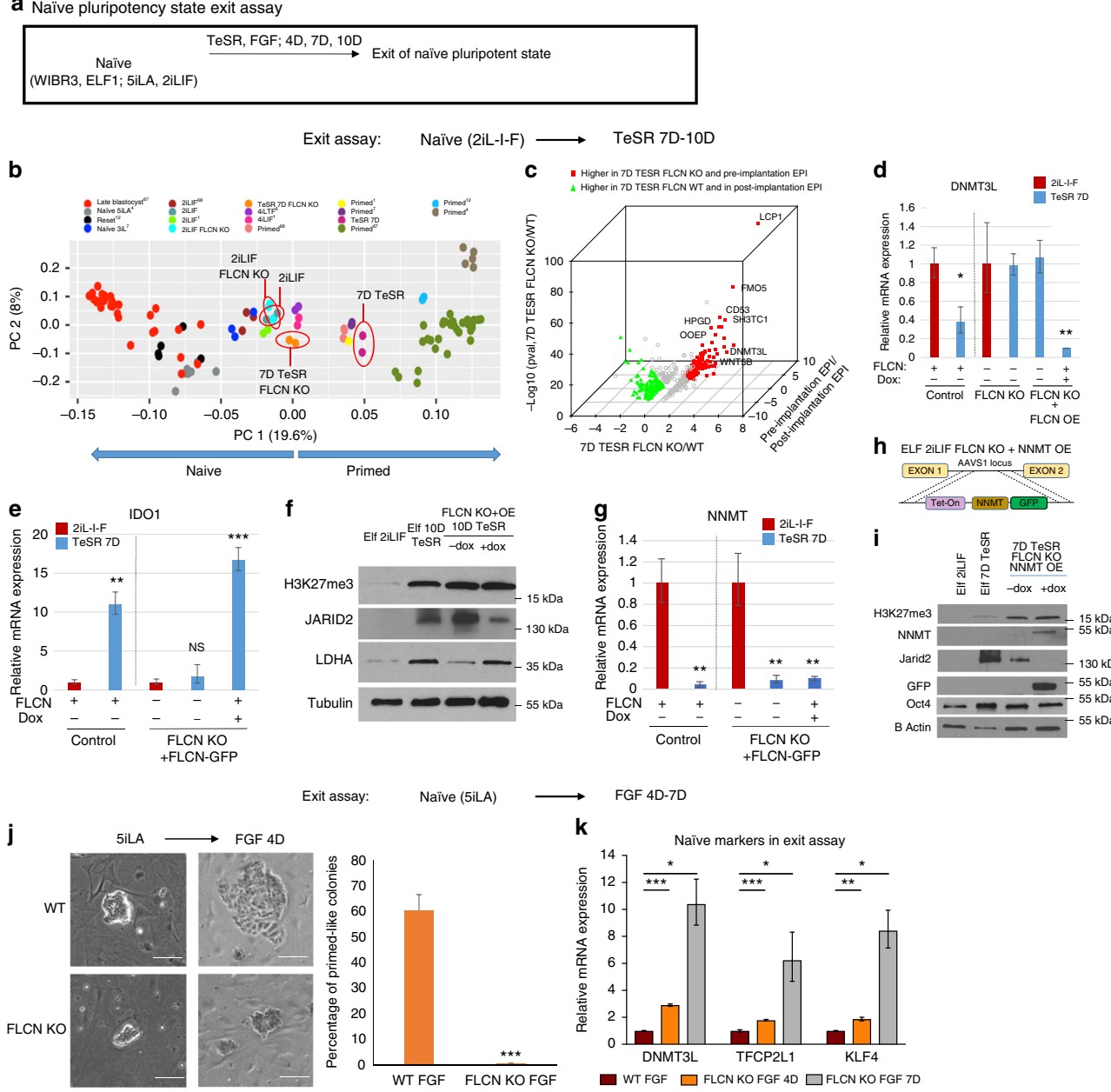

**Fig. 3** FLCN is essential for the exit from the naïve pluripotency. **a** Schematic representation of experimental procedures used to exit naïve pluripotent state. **b** Principal component analysis after RNA-seq analysis reveals that *FLCN* KO hESC do not fully exit naïve state. RNA-seq from various pre-implantation in vivo human embryo[67], naïve, and primed in vitro hESC were plotted, and separated in PC1 axis[1,6,7,11,12,67,68]. **c** Expression pattern of genes regulated by *FLCN* KO during exit of naïve state compared to cynomolgus monkey (*Macaca fascicularis*[34]) pre- vs. post-implantation stage. Red: genes up-regulated in 7D TeSR *FLCN* KO and pre-implantation; green: genes up-regulated in 7D TeSR WT and post-implantation. **d, e** RT-qPCR analysis of naïve (DNMT3L, **d**) and primed (IDO1, **e**) markers after exit of 2iL-I-F naïve state (7 day TeSR) in WT, FLCN KO, and rescue line (FLCN KO + OE FLCN-GFP). S.e.m.; *$p < 0.05$, **$p < 0.005$, ***$p < 0.001$; two-tailed *t*-test, $n = 3$–6 biological replicates. Presented are the fold changes compared to naïve 2iL-I-F. **f** Western blot analysis of primed markers JARID2, LDHA, and H3K27me3 marks in naïve 2iL-I-F hESC and 10D TeSR hESC. **g** RT-qPCR analysis of NNMT expression. S.e.m.; **$p < 0.005$; two-tailed *t*-test, $n = 3$–6 biological replicates. **h** Model of doxycycline inducible- NNMT-GFP fusion construct inserted into AAVS1 locus of 2iL-I-F FLCN KO hESC. **i** Western blot analysis of NNMT, GFP, H3K27me3, JARID2, and OCT4 after NNMT OE in FLCN KO 7D TeSR hESC. **j, k** FLCN KO cells retain naïve 5iLA morphology (**j**) and naïve markers DNMT3L, TFCP2L1, and KLF4 (**k**) when pushed to exit the naïve state (Exit assay: FGF 4 or 7 days; relative mRNA expression: fold changes of expression in FLCN KO vs. control (WT FGF: 4D or 6D, red bar). Complete data presented in Supplementary Fig. 3B. S.e.m.; *$p < 0.05$, **$p < 0.005$, ***$p < 0.001$; two-tailed *t*-test, $n = 3$–6 biological replicates. Scale bars represent 100 μm

line would alter these phenotypes (Fig. 3h, i). We induced the exit assay in the *FLCN* KO; NNMT OE hESC and tested marker expression. The primed marker, JARID2, was reduced due to NNMT overexpression in *FLCN* KO, suggesting that JARID2 is regulated by NNMT activity (Fig. 3i).

To test FLCN function in different naïve hESC conditions and genetic backgrounds, we analyzed the capacity of *FLCN* KO to exit hESC(5iLA) naïve conditions. When naïve hESC(5iLA) are induced to exit the naïve stage by eliminating 5iLA and growing the cells in the presence of bFGF (4–7 days), the control hESC

colonies downregulated naïve markers (Supplementary Fig. 3B) and lost the compact and mounded naïve hESC morphology (Fig. 3j). However, when the *FLCN* KO mutants were grown in this exit assay, the hESC colonies still exhibited compact and mounded hESC colony morphology and expressed higher levels of naïve markers DNMT3L, TFCP2L1, and KLF4 (Fig. 3j, k; Supplementary Fig. 3B). These data show that regardless of the naïve hESC growth conditions or genetic background, FLCN is required for the exit from this hESC pluripotent state.

**TFE3 localization is regulated by FLCN in hESC.** Analysis of transcription factor target enrichment in genes downregulated during hESC exit from the naïve state revealed TFE3 as an important regulator of the process (Supplementary Fig. 1C). Furthermore, characterization of transcription factor target enrichment among the genes upregulated in *FLCN* KO cells during the exit assay, also showed TFE3 among the top transcription factors (Fig. 4a). Combined, these results suggest that TFE3 based regulation of naïve exit was abnormal in *FLCN* KO hESC lines. We therefore investigated the potential of FLCN to regulate TFE3 function in hESC. The transcription factor TFE3 has previously been shown to accumulate in the cytoplasm in primed cells, both in mouse and human ESC[6,19]. We analyzed the localization of TFE3 in naïve (5iLA and 2iL-I-F) and primed hESC conditions (Fig. 4b–d). TFE3 localized in the nucleus in naïve 5iLA hESC, both in the nucleus and the cytoplasm in 2iL-I-F hESC, and mainly in the cytoplasm in primed cell lines and in cell lines exiting the naïve state (Fig. 4b–d). However, in *FLCN* KO lines TFE3 was localized exclusively in the nucleus, regardless of whether the cells were cultured in the naïve state or pushed to primed state (Fig. 4c, d). In the FLCN rescue line, TFE3 localization to the cytoplasm in cells exiting the naïve state was restored (Fig. 4c). These data show that in hESC, FLCN is required for regulation of cytoplasmic retention of TFE3.

Based on previously described post-translational regulation of TFE3 localization, we investigated whether FLCN can affect post-translational modification of TFE3[39,40]. In wild-type hESC, TFE3 molecular weight (MW) increases from 72 to 80 kDa, as cells exit the naïve pluripotent stage (Fig. 4e, 10D TeSR). This increase in TFE3 protein size is a result from phosphorylation of TFE3 and has been proposed to be responsible for the cytoplasmic retention of TFE3[39,40]. The phosphorylated TFE3 is missing in *FLCN* KO, suggesting that FLCN is required for this modification. However, we observed a higher molecular weight TFE3 protein (around 89–100 kDa) in *FLCN* KO hESC, consistent with previous reports in MEF and mouse kidney cells[40,41]. TFE3[89kDa] was not removed upon phosphatase treatment[40] and our RNAseq analysis revealed that only one TFE3 isoform corresponding to the 72 kDa band is expressed in naïve hESC and hESC exiting naïve state, suggesting that TFE3 undergoes another post-translational modification in *FLCN* KO hESC.

**WNT inhibition rescues the *FLCN* KO phenotype.** We demonstrated that when FLCN function is lost, naïve cells cannot develop to the primed state since the exit from naïve pluripotency is blocked by restriction of TFE3 localization to the nucleus (Figs. 3 and 4c, d). TFE3 has been shown to regulate the transcription of the core pluripotency factor Esrrb in mouse ESC[19]. However, even though ESRRB is an essential factor in mouse ground state[19,20,37,42], RNAseq data suggest that it is expressed at a very low level in stabilized hESC lines (Supplementary Fig. 4A) and in the pre-implantation epiblast of cynomolgus monkey (see ref.[33], Supplementary Fig. 4B). Interestingly, ESRRB expression is observed in early ICM of cynomolgus monkey in vivo (Supplementary Fig. 4B)[33]. We analyzed ESRRB expression by RT-qPCR

and western blot and were able to detect ESRRB transcript and protein in hESC (Fig. 4g; Supplementary Fig. 4C, D). Of the differentially expressed genes in naïve 2iL-I-F *FLCN* KO hESC vs. naïve 2iL-I-F wild-type hESC, 672 out of 1787 upregulated genes are TFE3 targets (overlap hypergeometric test *p*-value $7.01 \times 10^{-126}$; Fig. 4f, Supplementary Data 3A), including ESRRB, TFCP2L1, and members of the Wnt pathway (Fig. 4f; see ref.[20]). *FLCN* KO up-regulates ESRRB expression in the naïve state but not upon exit from human naïve pluripotency (Fig. 4f, i; Supplementary Data 3A and 4A; Supplementary Fig. 4C, D), suggesting that ESRRB is not responsible for the *FLCN* KO phenotype. Furthermore, *FLCN;ESRRB* double KO or *FLCN* KO; *ESRRB* mutant are not able to rescue the *FLCN* KO mutant phenotypes in transition assays (Fig. 4g, h; Supplementary Fig. 4E–H). ESRRB reduction does not upregulate the primed markers HIF1α and LDHA or downregulate the naïve TFCP2L1, DNMT3L, and KLF4. These data suggest that TFE3-dependent overexpression of ESRRB cannot be causal for the lack of *FLCN* KO to exit the 5iLA and 2iL-I-F states. Knock-down (KD) of TFCP2L1 with a lentivirus expressing shRNAs targeting TFCP2L1 slightly decreased KLF4 expression but did not affect DNMT3L expression in *FLCN* KO exiting the naive 5iLA state, suggesting that TFCP2L1 plays a minor role on FLCN phenotype (Supplementary Fig. 4I–L). KLF4 is also upregulated in *FLCN* KO and KLF4 KD with shRNA partially decreases naïve hESC markers DNMT3L and TFCP2L1. However, since KLF4 is not a predicted TFE3 target, KLF4 cannot explain the mechanism of FLCN function during the exit from naïve pluripotency (Supplementary Fig. 4I–L).

To identify the genes in *FLCN* KO cells that are causal for the defective exit from the naïve state we analyzed differentially expressed genes in a transition assay; in TeSR 7D 673 out of 1952 upregulated genes in *FLCN* KO hESC vs. wild-type hESC were TFE3 targets (overlap hypergeometric test *p*-value $2 \times 10^{-69}$; Fig. 4i; Supplementary Data 4D). As seen in naïve state (Fig. 4f) among these TFE3 target genes were several members of the WNT pathway, including WNT ligands WNT3A, WNT5B, WNT6, and WNT11. *FLCN* KO also resulted in up-regulation of WNT targets during the exit form naïve pluripotency (Fig. 4i–l). We tested if WNT pathway up-regulation in *FLCN* KO was causal for the inability to exit from naïve pluripotency by using an inhibitor of WNT processing and secretion, IWP2 and analyzing genome wide gene expression changes by RNA seq. Principal component analysis of the gene expression changes revealed that Wnt inhibition in *FLCN* KO cells rescued the defect in exit from human naïve pluripotency (Fig. 4m), allowing up-regulation of primed markers and down-regulation of naïve markers, as seen in wild-type hESC exiting the naïve state (Supplementary Fig. 5A–C). The data suggest that TFE3-dependent WNT expression blocks the exit from the naïve state (5iLA and 2iL-I-F) in *FLCN* KO hESC.

**FLCN acts through mTORC1 and mTORC2.** To reveal the mechanism of FLCN function in pluripotency, we investigated why FLCN expression is critical during exit from the naïve pluripotent stage. FLCN expression levels are similar in naïve and primed hESC lines in vitro or in cynomolgus monkey pre- and post-implantation stages in vivo (Supplementary Fig. 5D, E). We hypothesized that FLCN could form different complexes at different pluripotency states. In order to uncover the proteins interacting with FLCN, we performed immunoprecipitation (IP) of FLCN-GFP in *FLCN* KO cells, followed by mass spectrometry analysis in naïve cells and cells exiting the naïve state (3D TeSR). We found that FLCN is associated with its known interactors FNIP1 and FNIP2, as well as desmosome proteins, HDAC6 and Caspase 14, among others (Fig. 5a,

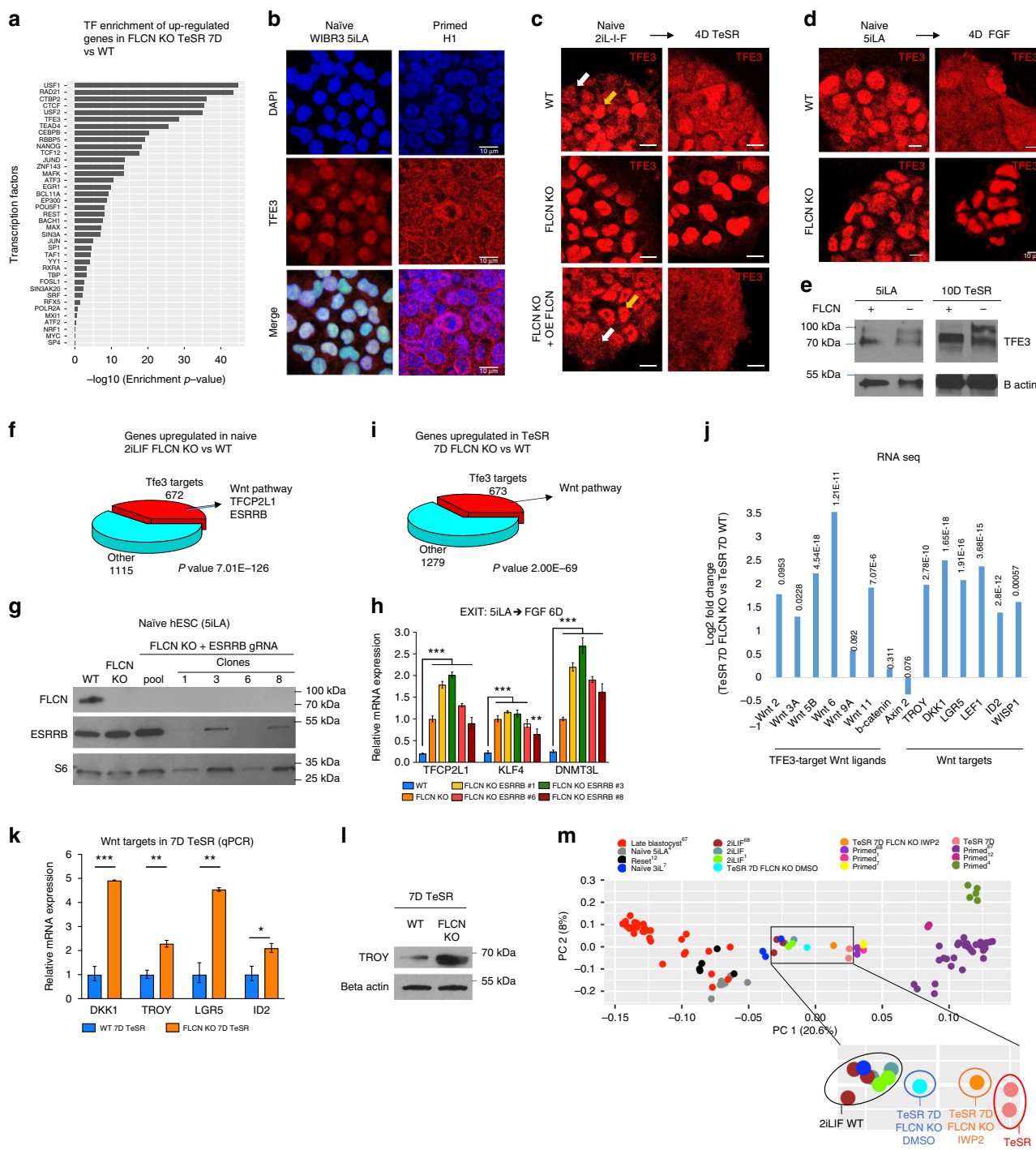

**Fig. 4** FLCN regulates TFE3 subcellular localization in hESC. **a** Enrichment of transcription factor targets in genes that are up-regulated in FLCN KO 7D TeSR compared to WT 7D TeSR. **b** Immunofluorescence staining of TFE3 in naïve (WIBR3 5iLA) and primed (H1) hESC observed with confocal microscopy. **c**, **d** WT and FLCN KO lines were stained for TFE3 in naïve conditions (2iL-I-F, C; 5iLA, D) and 4 days after culture in TeSR (**c**) or FGF (**d**) media. TFE3 was detected in the nucleus in WT naïve hESC and in FLCN KO grown in all conditions. Scale bars represent 10 μm. **e** Western blot analysis of TFE3 in WT and FLCN KO in naïve (5iLA) and 10D TeSR samples. **f** TFE3 targets are upregulated in naïve hESC (2iL-I-F) FLCN KO compared to WT, including TFCP2L1, ESRRB, and Wnt pathway components. **g** Generation of FLCN/ESRRB double KO in naïve hESC. Western blot analysis of CRISPR-Cas9-generated ESRRB mutant clones in WIBR3 5iLA FLCN KO. **h** FLCN/ESRRB double mutant does not rescue FLCN KO phenotype during the exit of naïve pluripotency. RT-qPCR analysis of naive hESC markers (TFCP2L1, KLF4, and DNMT3L) in WIBR3 6D FGF (WT), WIBR3 FLCN KO 6D FGF, and WIBR3 FLCN KO/ESRRB mutants 6D FGF. S.e.m.; **$p < 0.005$, ***$p < 0.001$; two-tailed $t$-test. **i** TFE3 targets are upregulated in 7D TeSR FLCN KO hESC compared to 7D TeSR WT hESC, including several Wnt pathway components. **j** TFE3-target Wnt ligands and Wnt pathway targets are up-regulated in naïve (2iL-I-F) hESC FLCN KO compared to WT (log2 fold change from RNAseq data is presented and $p$-value indicated). **k**, **l** Wnt-pathway targets are upregulated in 7D TeSR FLCN KO hESC compared to 7D TeSR WT hESC, as analyzed by RT-qPCR (**k**, s.e.m.; *$p < 0.05$,**$p < 0.005$, ***$p < 0.001$; two-tailed $t$-test, $n = 3$ biological replicates) and western blot (**l**). **m** Principal component analysis after RNA-seq analysis reveals that inhibition of Wnt by IWP2 during the exit of naïve state (Elf1 7D TeSR) rescues FLCN KO phenotype. RNA-seq from various pre-implantation in vivo human embryo[67], naïve, and primed in vitro hESC were plotted, and separated in PC1 axis[1,6,7,11,12,67,68]

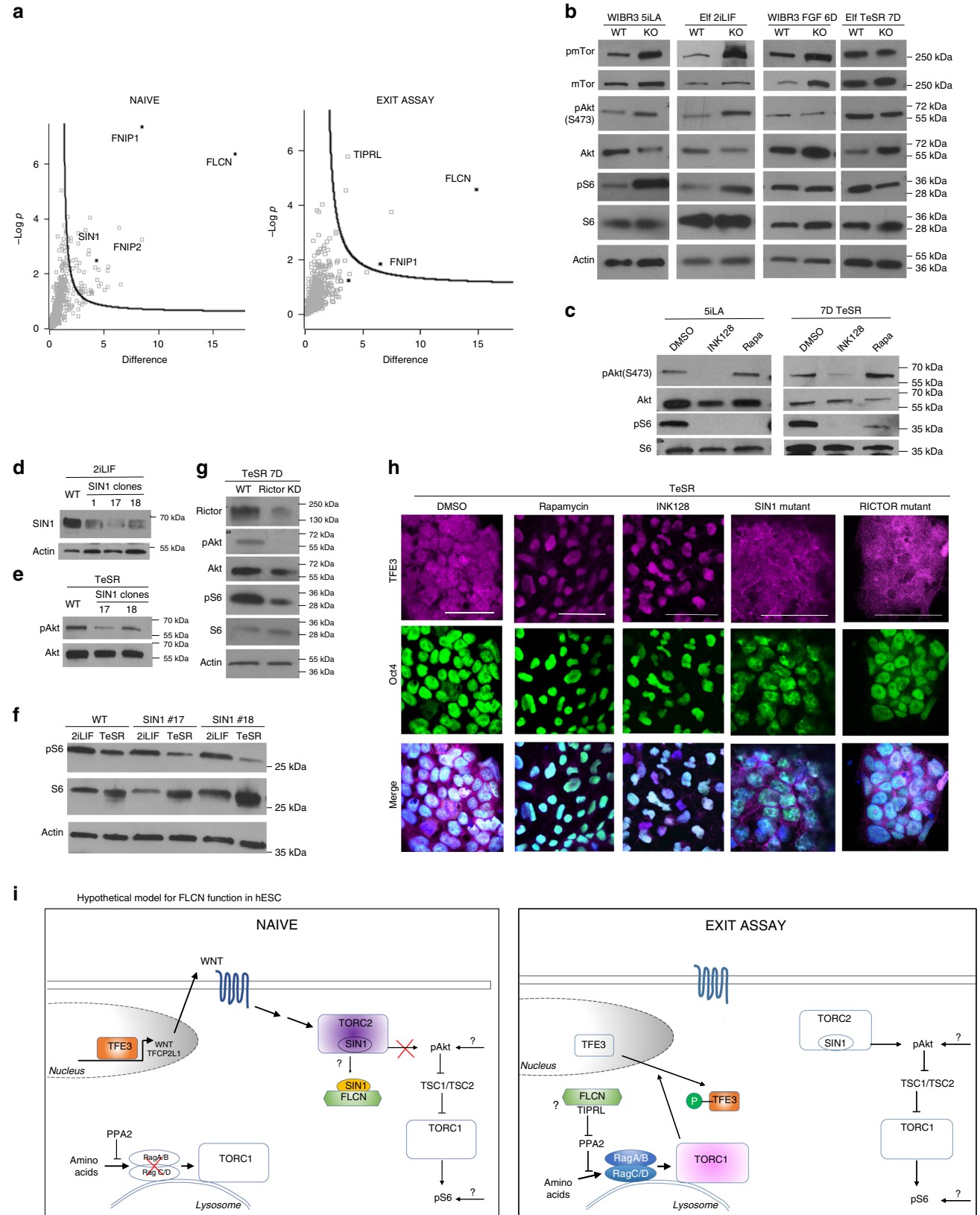

Supplementary Data 5). Interestingly the mTORC2 component MAPKAP1(SIN1) was found in a complex with FLCN only in naïve hESC while TIPRL, ARAP1, RAP2B, CD59, and PLCH1 were found in a complex with FLCN only in cells exiting the naïve state (Fig. 5a; Supplementary Data 5). We validated the specific binding of SIN1 to FLCN in naïve hESC by IP of

FLCN-GFP followed by western blot analysis (Supplementary Fig. 6A).

Since FLCN and several of its new binding interactors have been previously associated with mTOR pathways[30,31], we assessed the phosphorylation status of mTor as well as mTORC1 and mTORC2 targets (S6 and AKT(S473), respectively) in hESC

**Fig. 5** FLCN differentially regulates mTOR pathway in various pluripotency states. **a** Proteins associated with FLCN were co-immunoprecipitated and identified using mass spectrometry. Protein abundance (label free quantification (LFQ) intensity) differences were quantified between plus and minus dox-treated cell lines. Black line represents the 5% false discovery rate calculated in Perseus 1.5.6. $N = 3$–4 technical measurements. **b** Western blot analysis of p-mTor, mTOR, p-AKT(Ser 473), AKT, p-S6, and S6 in WT or *FLCN* KO naïve hESC (5iLA, 2iL-I-F) and hESC growing in FGF (7D) or TeSR (7D). **c** Effect of Rapamycin and INK-128 on naïve hESC and hESC exiting the naïve pluripotent state. Naïve 5iLA and 7D TeSR hESC were treated with either DMSO (control), INK-128 (100 nM), or Rapamycin (100 nM) for 24 h (at day 6 for 7D TeSR cells). **d** Western blot analysis of SIN1 expression in naïve (2iL-I-F) hESC WT and SIN1 mutant lines. **e, f** SIN1 mutant hESC exhibit a reduction of p(S473)AKT (**e**) and pS6 (**f**) in cells exiting the naïve state (TeSR 7D). **g** RICTOR mutant hESC exhibit a reduction of p(S473)AKT and pS6 in cells exiting the naïve state (TeSR 7D). **h** TFE3 is observed in the cytoplasm of hESC exiting the naïve state (Elf1 2iL-I-F → 4D TeSR, WT, SIN1 mutant clone #18, RICTOR KD mutant) and in the nucleus when treated for 24 h with mTORC1 inhibitor Rapamycin (100 nM) or mTORC1/2 inhibitor INK-128 (100 nM), using confocal microscopy. Scale bars represent 50 μm. **i** Hypothetical model of FLCN-regulation of mTORC1, mTORC2, and TFE3 localization in hESC

grown in various conditions (5iLA, 2iL-I-F, FGF, and TeSR). *FLCN* KO resulted in increased phosphorylation of mTOR, S6, and AKT(S473) in naïve cells (5iLA and 2iL-I-F) but not in cells exiting the naïve state (FGF and TeSR) (Fig. 5b; Supplementary Fig. 6B). Interestingly, FLCN does not seem to affect the phosphorylation of AKT on T308 (Supplementary Fig. 6C). These data suggest that FLCN inhibits mTORC2 and mTORC1 in the naïve state.

To analyze the mTORC2 function in naïve-to-primed transition we generated mutant alleles for SIN1 (MAPKAP1) in hESC (Fig. 5d). While no obvious defects were observed in naïve SIN1 mutant hESC (Fig. 5f; Supplementary Fig. 6D), during primed transition mTORC2 target, P-AKT(S473) was downregulated (Fig. 5e). Interestingly, mTORC1 target P-S6 was downregulated in SIN1 mutants, suggesting that a branch of mTORC1 is downstream of mTORC2 in primed hESC (Fig. 5f). mTORC2-independent roles of SIN1 have been described[43,44]. We therefore generated KD of another member of mTORC2, RICTOR in hESC and found that a decrease of RICTOR expression also downregulated P-AKT(S473) and P-S6 during primed transition, mimicking SIN1 mutant phenotype in cells exiting the naïve state (Fig. 5g).

We then analyzed whether mTORC1 and/or mTORC2 affect the localization of TFE3. We treated hESC exiting the naïve state with a mTORC1/2 inhibitor, INK-128 and mTORC1-specific inhibitor Rapamycin (Fig. 5c, h). While TFE3 exited the nucleus in control hESC in the primed stage, TFE3 did not exit the nucleus after cells were treated with INK-128 or Rapamycin (Fig. 5h), suggesting that TFE3 localization in hESC is controlled by mTOR activity. To analyze mTORC2 function in TFE3 localization we tested TFE3 localization in the SIN1 and RICTOR mutants. We show that mTORC2 is dispensable for TFE3 release from the nucleus since TFE3 exits the nucleus in SIN1 and RICTOR mutants during hESC naïve-to-primed transition (Fig. 5h). The specific inhibition of mTORC1 by Rapamycin, evidenced by decrease of P-S6 and P-4EBP1, but not P-AKT (S473), represses TFE3 exit from nucleus in the naïve-to-primed transition assay (Fig. 5c, h; Supplementary Fig. 6F–H). These data suggest that mTORC1, but not mTORC2, is required for TFE3 exit from the nucleus in hESC primed stage (Fig. 5h).

mTOR requirement in primed hESC was further reciprocated with a small-molecule screen. We tested compound sensitivity profiles of naïve and primed hESC in a high throughput screen of 160 approved and investigational oncology drugs (Supplementary Fig. 6E). Compounds were added at concentrations ranging from 5 pM to 100 μM using the CyBio CyBi-Well Vario and incubated at 37 °C, 5% $CO_2$ for 72 h. Viability was assessed using quantitation of luminescence derived from intracellular ATP. The most significant new set of small molecules that reduced primed but not naïve hESC viability was mTOR/PI3K/AKT inhibitors (AZD-8055, BEZ-235, BGT-226, BKM-120, Everolimus, GDC-0941, OSI-027, PF-04691502, PKI-587, Pp-242, Rapamycin, Temsirolimus), suggesting that both mTORC1 and mTORC2 are critical for primed hESC stage.

Similar results have been obtained recently using haploid cells[45].

## Discussion

Our CRISPR-Cas9 KO screen led to identification of new regulators of the exit from naïve pluripotency in humans, including FLCN. We showed that FLCN controls the localization of the transcription factor TFE3 through the mTOR pathway. Localization of TFE3 in the nucleus results in the transcriptional activation of the WNT pathway, thereby retaining the hESC in the naïve stage. Inhibiting the WNT pathway rescues the capacity of a *FLCN* KO hESC line to exit naïve pluripotency, suggesting that a key TFE3 target, regulated by FLCN in hESC is the WNT pathway.

The tumor suppressor, cytoplasmic guanine exchange factor FLCN, is associated with Birt–Hogg Dubé syndrome, which is characterized by multiple benign hair follicle tumors, renal and pulmonary cysts, and increased susceptibility to kidney cancer[30]. Interestingly, Gene Ontology analysis reveals that genes involved in regulation of kidney development are enriched in *FLCN* KO vs. *FLCN* KO + FLCN-GFP rescue (Supplementary Fig. 2E). FLCN has been shown to regulate various signaling pathways and cellular metabolism through mTOR, AMPK, HIF1α, ciliogenesis, autophagy, and lysosomal biogenesis[30,46]. It is an essential protein required for *Drosophila* germline stem cell maintenance and hematopoietic stem cell maintenance[31,47,48]. However, Flcn is not required for the maintenance of mouse ESC[19,31] or naïve hESC (this study), suggesting that Flcn function is context dependent. Mouse *Flcn*−/− embryos die soon after implantation, and Flcn KD mouse ESC loose the capacity to differentiate[19,31]. This phenotype is at least partly due to the fact that FLCN controls the nuclear localization of a key stem cell transcription factor, TFE3 (refs. [19,40,47]).

Recent findings have revealed that FLCN acts at the lysosomal membrane, interacting with Rag GTPase[49,50]. This interaction regulates amino acid-dependent activation of mTOR. Furthermore, amino acid induced action of FLCN as a guanine nucleotide exchange factor for RagA/B-complex[51] and a GTPase-activating protein for RagC/D[49] activates Rag and mTOR, which could phosphorylate MiT-TFE transcription factors, allowing them to move out of the nucleus[39]. Although TFE3 is localized in the nucleus in naïve hESC but in the cytoplasm in primed hESC or hESC exiting the naïve state, this localization difference cannot be explained by the level of expression of FLCN, since FLCN is expressed at the same level in various pluripotent states. FLCN has been shown to interact with FLCN Interacting Proteins 1 and 2 (FNIP1/2)[31,47,52]. We showed by co-immunoprecipitation of FLCN-GFP followed by mass spectrometry analysis that FLCN also binds to FNIP1/2 in hESC. In addition, FLCN binds to TOR Signaling Pathway Regulator (TIPRL) upon exiting from the naïve state but not in the naïve state, i.e., mTOR signaling pathway mechanics change as cells progress from naïve to

primed. TIPRL has been shown to enhance amino acid-dependent mTORC1 signaling by inhibiting the phosphatase activity of PP2Ac[53]. We propose that the formation of a FLCN/FNIP1/TIPRL complex in cells exiting the naïve state could activate mTOR at the lysosome (through Rags) and induce phosphorylation of TFE3 resulting in its retention in cytoplasm (Fig. 5i).

In cancer cells FLCN mutants exhibit activation of both mTORC1 and mTORC2 and treatment of a mouse model of Flcn mutant-induced renal carcinoma with mTor inhibitor reduces tumor size[31]. However, the mechanism of how FLCN controls both mTORC1 and mTORC2 function is unknown. It has been proposed that naïve and primed hESC are differentially dependent on mTORC2 (ref. [54]). Here we show that FLCN can bind to SIN1, an essential component of mTORC2 (ref. [55]) only in the naïve state. Using SIN1 and RICTOR-specific mutant we show that while mTORC1 is important for TFE3 localization, mTORC2 is dispensable for the process. Reduction of either SIN1 or RICTOR expression decreased the levels of P-S6, in addition to P-AKT(S473), suggesting that mTORC2 can partially regulate mTORC1 in hESC, as seen previously in mESC[56]. Future experiments will reveal the function of this branch of mTORC1 regulation. We propose that FLCN/SIN1 interaction may negatively regulate mTORC2 activity, and consequently cytoplasmic mTORC1 activity (Fig. 5i). However, we cannot rule out that the binding of FLCN to SIN1 in naïve hESC could also regulate mTORC2-independent functions of SIN1 (refs. [43,44]). It is possible that FLCN may not directly interact with mTORC2 but rather may sequester SIN1 from this complex. The role of mTORC2 and cytoplasmic mTORC1 in the exit from naïve pluripotency remains to be identified. Interestingly we also found that during the exit of naïve pluripotency FLCN binds to ARAP1, RAP2B, CD59, and PLCH1, four proteins involved in the EGFR signaling pathway, corroborating the recent findings that FLCN negatively regulates EGFR signaling[57]. The role of FLCN on EGFR pathway in hESC will be interesting to examine further in future studies, in particular its role upstream of AKT and mTOR pathway.

In human ESC, we observed that in the absence of FLCN, TFE3 is exclusively localized to the nucleus, resulting in the activation of its target genes. TFE3 has been shown to regulate the transcription of the core pluripotency factor Esrrb in mouse ESC[19]. However, ESRRB expression is low in hESC lines[13,58], suggesting that TFE3 targets are different in human and mouse. KO of ESRRB in naïve FLCN KO hESC does not significantly decrease the expression of naïve markers during the naïve to primed transition. Among the genes up-regulated in FLCN KO hESC were several TFE3-targets WNT ligands, as well as WNT pathway targets, indicating WNT pathway activation. WNT plays an essential role in maintaining the naïve pluripotent state in mouse and human ESC[1,20,59,60]. Since the expression of Wnt has been shown to block the transition from naïve to primed ESC[60], we hypothesized that activation of the WNT pathway by TFE3 in FLCN KO hESC prevents the cells from exiting the naïve state. We confirmed that hypothesis by showing that WNT inhibitor was able to rescue the FLCN KO phenotype. These studies may have bearing for novel treatments of human cancer caused by Folliculin mutations.

## Methods

**hESC culture.** Naïve hESC [Elf-1 (NIH_hESC Registry #0156), Elm-2 (NIH_hESC Registry #0396), Elf-3 (NIH_hESC Registry #0397), Elf-4 (NIH_hESC Registry #0398), and WIBR3 (NIH registry#0079) toggled in 5iLA] were cultured as previously described[1,3,4,32]. Briefly, cells were grown on a feeder layer of irradiated primary mouse embryonic fibroblasts (MEF) in naïve hESC media (2iL-I-F or 5iLA). 2iL-I-F media consisted of DMEM/F-12 media supplemented with 20% knockout serum replacer (KSR), 0.1 mM nonessential amino acids (NEAA), 1 mM sodium pyruvate, penicillin/streptomycin (all from Invitrogen, Carlsbad, CA), 0.1 mM β-mercaptoethanol (Sigma-Aldrich, St. Louis, MO), 1 μM GSK3 inhibitor (CHIR99021, Selleckchem), 1 μM of MEK inhibitor (PD0325901, Selleckchem), 10 ng ml$^{-1}$ human LIF (Chemicon), 5 ng ml$^{-1}$ IGF1 (Peprotech) and 10 ng ml$^{-1}$ bFGF. 5iLA media consisted of 50:50 mixture of DMEM/F-12 (Invitrogen; 11320) and Neurobasal (Invitrogen; 21103) media, supplemented with 1% N2 supplement (Invitrogen; 17502048), 2% B27 supplement (Invitrogen; 17504044), 1 mM glutamine (Invitrogen), 1% nonessential amino acids (Invitrogen), 0.1 mM β-mercaptoethanol (Sigma), penicillin–streptomycin (Invitrogen), 50 mg ml$^{-1}$ bovine serum albumin (BSA) (Sigma), BRAF inhibitor (0.5 μM), SRC inhibitor (1 μM), MEK inhibitor (1 μM), GSK3 inhibitor (IM-12, 1 μM), ROCK inhibitor (10 μM), recombinant human LIF (20 ng.ml$^{-1}$), and Activin A (10 ng ml$^{-1}$). Cells were passaged using 0.05% Trypsin-EDTA (Life Technologies). Cells were transferred to matrigel-coated plates prior molecular analysis. Exit from the naïve state was achieved by culturing the cells in either mTeSR1 media (StemCell Technologies) or primed FGF hESC media (DMEM/F-12 media supplemented with 20% KSR, 0.1 mM NEAA, 1 mM sodium pyruvate, penicillin/streptomycin, 0.1 mM β-mercaptoethanol, and 10 ng ml$^{-1}$ bFGF) on matrigel-coated plates. All cells were cultured at 5% $O_2$ and 5% $CO_2$ between passage p20 and p40, as indicated.

**Whole-genome CRISPR screen during exit from naïve state.** Ten to 20 million naïve hESC Elf1 2iL-I-F (passage p21–p26) were transduced with human pooled lentiCRISPR library (genome-scale CRISPR-Cas9 knockout (GeCKO) library[25]) via spinfection and plated onto irradiated Drug Resistance 4 (DR4) MEF. Four days later, puromycin was added (0.5 μg ml$^{-1}$) for 2 days. Cells were passaged at day 7 and day 10. On day 10, cells were switched to TeSR media to initiate the exit from the naïve state. After 4 days in mTeSR1, cells were treated with Methotrexate (1 μM; Torcis) and Acetaldehyde (1 mM; Sigma) (M/A). In all, 3–4 × 5 millions cells were harvested at Day 14 time point (before M/A selection) and at Day 17 (3 days after beginning of M/A selection). Genomic DNA was extracted and sgRNA sequences were PCR amplified and sequenced. Bowtie[61] was used to align the sequenced reads to the sgRNA library and the R/Bioconductor package edgeR[62,63] with voom normalization[64] was used to assess changes across various groups. Guides having a fold change greater than 1 and an adjusted false discovery rate (FDR) >0.05 were considered statistically significant. Raw and mapped data files are available at the Gene Expression Omnibus database (GSE).

**Generation of CRISPR mutant lines.** Naïve hESC (Elf1 2iL-I-F or WIBR3 5iLA) were spin-infected with lentiCRISPR-v2 lentiviral pools of four sgRNA per gene for FLCN and ESRRB in the presence of 4 μg ml$^{-1}$ polybrene (Supplementary Data 6A) and plated onto irradiated DR4 MEF. Two days later, cells were selected with puromycin (0.5 μg ml$^{-1}$, for 2–3 days). Alternatively, SIN1 and RICTOR mutants were generated in Elf1-iCas9 (ref. [1]). Guides were ordered through Synthego (RICTOR) or as T7-gRNA primers (SIN1). A dsDNA fragment was synthesized from these primers by self-annealing PCR to a complementary scaffold primer. The dsDNA fragment was followed by Q5 High-Fidelity-based PCR (New England Biolabs). This 120 bp strand served as a template for IVT (MAXIscript T7 kit; Applied Biosystems). The RNA was then purified using Pellet Paint® Co-Precipitant (Novagen). Elf1-iCas9 cells were treated with doxycycline (2 μg ml$^{-1}$) for 2–3 days before and during transfection. Cells were transfected with 40 nM of gRNA using Lipofectamine RNAiMAX (Life Technologies). A second transfection was performed after 24 h. Two days after the last gRNA transfection, Elf1-iCas9 cells were dissociated into single cells and replated onto MEF-coated plates or collected for DNA analysis (pool). In the next passage, single colonies from MEF-coated plates were randomly selected and amplified. The molecular characterization of the mutant lines and their phenotype were analyzed between passage 33 and 40 using controls with matching passage number.

Genomic DNA was extracted using DNAzol reagent (Invitrogen) according to the manufacturer's instructions and quantified using Nanodrop ND-1000. Genomic regions flanking the CRISPR target sites were PCR amplified with the designed primers (Supplementary Data 6C) and sent for Sanger sequencing. Alternatively, PCR were purified with the PureLink Quick PCR Purification Kit (Invitrogen), inserted into a carrier vector by Gibson Assembly, and 96 clones Sanger sequenced.

**Secondary CRISPR screen.** Newly generated naïve hESC (Elf1 2iL-I-F) CRISPR KO were pushed to exit the naïve pluripotent state by switching their media with mTeSR1. Four days later, cells were treated with methotrexate (1 μM) and acetaldehyde (1 mM) for three additional days. Number of surviving cells was then assessed using NucleoCounter NC-200 (ChemoMetec).

**Generation of FLCN and NNMT overexpression constructs.** NNMT gene was amplified from the pCLNCX-NNMT OE construct[65] using PCR with Q5 High-Fidelity DNA Polymerase (New England Biolabs). PCR product was Gibson assembled into mammalian vector AAVS1-TRE3G-EGFP[66]. Full-length FLCN was amplified out of a cDNA library and inserted into the MluI site of AAVS1-TRE3G-EGFP by Gibson Assembly. 1 × 10$^6$ cells of naïve FLCN KO hESC (Elf1 2iL-I-F) were transfected with 0.5 μg AAVS1-TALEN-R plasmid (Addgene #59026), 0.5 μg AAVS1-TALEN-L (Addgene #59025), and 4 μg of either AAVS1-TRE3G-NNMT-

EGFP or AAVS1-TRE3G-FLCN-EGFP using Amaxa Lonza Human stem cell Kit #2. The cells were then plated with 5 µM ROCK inhibitor onto irradiated DR4 MEF. Two days following the nucleofection, the cells were selected for Puromycin 0.5 µg ml⁻¹ for 2 days.

**Generation of KLF4 and TFCP2L1 knock-down lines using shRNA**. shRNA against TFCP2L1 or KLF4 were inserted in the lentiviral backbone pLL3.7-EF1a-GFP (shRNA-GFP). WIBR3 5iLA FLCN KO cells were infected with pooled TFCP2L1 or KLF4 shRNA-GFP virus in the presence of polybrene (4 µg ml⁻¹ polybrene). The GFP expression was used to measure transfection efficiency. The knock-down efficiency of the shRNA was assessed by RT-qPCR and western blot analysis. Sequences of shRNA used are presented in Supplementary Data 6B.

**ChIP-seq experiment**. Naïve 2iL-I-F hESCs Elf1 NNMT KO (mutant 6.2.4; ref. [1]) were crosslinked and chromatin processed as previously described[1] with minor modifications. Briefly, cells were harvested with accutase and crosslinked in suspension with 1% formaldehyde solution for 10 min at room temperature. Reaction was quenched with glycine and crosslinked cells were rinsed with ice-cold phosphate-buffered saline (PBS). Nuclei were isolated and chromatin sonicated using a Covaris E210 to approximately 200–500 bp size range. Magnetic Dynabeads were incubated overnight rotating at 4 °C with antibody against H3K27me3 (Active Motif, cat # 39155). Sonicated chromatin from approximately 200 thousand to 3 million cells was added to the bead-bound-antibodies and allowed to incubate at 4 °C rotating overnight. Beads were washed to remove unbound chromatin. Bound chromatin was eluted from beads and reverse crosslinked overnight. Purified DNA was prepared for next-generation sequencing via end repair, A-tailing, ligation of custom Y-adapters, and PCR amplification to generate final DNA library following gel size selection.

**WNT and mTORC1/2 chemical inhibition**. Wnt secretion was inhibited in hESC Elf1 and WIBR3 during the exit of the naïve state (7 days in either mTeSR1 or FGF, respectively) by treatment with IWP2 (7 days, 2 µM; Tocris). Dimethyl sulfoxide (DMSO) treatment was used as a vehicle control. mTORC1/2 activity was inhibited in hESC Elf1 during the exit of naïve state (4 days in mTeSR1) by treatment with INK-128 (100 nM; Medchem) or Rapamycin (100 nM) for 24 h (added 6 days after mTeSR1).

**Immunofluorescence staining**. Cells were fixed in 4% paraformaldehyde in PBS for 10 min, permeabilized for 10 min in 0.1% Triton X-100, and blocked for 1 h in 2% BSA. The cells were then incubated in primary antibody overnight, washed with PBS (3 × 5 min), incubated with the secondary antibody in 2% BSA for 1 h, washed (3 × 10 min) and stained with 1 µg ml⁻¹ DAPI for 10 min. Mounting media was composed of 2% of n-propyl gallate in 90% Glycerol and 10% PBS. Analysis was done on a Leica TCS-SPE Confocal microscope using a ×40 objective and Leica Software. The antibodies used for immunostaining were anti-TFE3 (Sigma Prestige HPA023881, 1:400), anti-Oct4 (Santa Cruz sc-5279, 1:100), and Alexa 488- or Alexa 647-conjugated secondary antibodies (Molecular Probes).

**Western blots analysis**. Cells were lysed directly on the plate with a lysis buffer containing 20 mM Tris-HCl pH 7.5, 150 mM NaCl, 15% glycerol, 1% Triton x-100, 1 M β-glycerolphosphate, 0.5 M NaF, 0.1 M sodium pyrophosphate, orthovanadate, PMSF, and 2% sodium dodecyl sulfate (SDS). Twenty-five units of Benzonase® Nuclease (EMD Chemicals, Gibbstown, NJ) was added to the lysis buffer right before use. Proteins were quantified by Bradford assay (Bio-rad), using BSA as Standard using the EnWallac Vision. The protein samples were combined with the 4× Laemmli sample buffer, heated (95 °C, 5 min), and run on SDS-PAGE (protean TGX pre-casted 7.5% gel or 4–20% gradient gel; Bio-rad) and transferred to the nitro-cellulose membrane (Bio-Rad) by semi-dry transfer (Bio-Rad). Membranes were blocked for 1 h with 5% milk or 2% BSA (for antibodies detecting phosphorylated proteins), and incubated in the primary antibodies overnight at 4 °C. The antibodies used for western blot were β-tubulin III (Promega G7121, 1:1000), β-actin (Cell Signaling 4970, 1:10000), Oct4 (Santa Cruz sc-5279, 1:1000), H3K27me3 (Active Motive 39155, 1:1000), FLCN (Cell Signaling D14G9, 1:2000), NNMT (Abcam 58743, 1:500), TFE3 (Sigma Prestige HPA023881, 1:1000), JARID2 (Cell Signaling D6M9X, 1:1000), LDHA (Cell Signaling 2002, 1:1000), SIN1 (EMD Millipore 05-1044, 1:000), pAkt (S473) (Cell Signaling 9271, 1:1000), pAkt(T308) (Cell Signaling 9275, 1:1000), Akt (Cell Signaling 9272, 1:1000), pS6 (Cell Signaling 2215, 1:10,000), S6 (Cell Signaling 2117, 1:2000), p4EBP1 (Cell Signaling 236B4, 1:1000), ESRRB (ProteinTech, 1:1000), pmTOR (Cell Signaling 2971, 1:1000), mTOR (Cell Signaling 2972, 1:1000), RICTOR (Invitrogen MA5-15681, 1:1000), TFCP2L1 (Novus Biologicals NBP1-85441, 1:1000), and KLF4 (Abcam ab129473, 1:1000). The membranes were then incubated with secondary antibodies (1:10,000, goat anti-rabbit or goat anti-mouse IgG HRP conjugate(Bio-Rad) for 1 h and the detection was performed using the immobilon-luminol reagent assay (EMD Millipore). Uncropped scans of the most important blots are presented in Supplementary Data 7.

**Co-immunoprecipitation**. Elf1 FLCN KO+FLCN-GFP hESC grown in either naïve media (2iL-I-F) or mTeSR1 media (3–4 days) plus or minus doxycycline (1 µg ml⁻¹) were incubated directly on the plate with 200 µl lysis buffer (10 mM Tris-HCl pH 7.5, 150 mM NaCl, 0.5 mM EDTA, 1.5 mM MgCl₂, 1 mM DTT, 25 mM NaF, 5% glycerol, 0.5% NP-40, (1 tablet/10 ml) PMSF added freshly). The lysate was transferred to a microtube and incubated on ice for 30 min with extensive vortexing every 10 min. Cell lysate was centrifuged at 15,600 × g for 15 min at 4 °C. One hundred and fifty microliters supernatant were incubated for 1 h at 4 °C with 20 µl of GFP-Trap-A beads (gta-20; Chromotek) used as per the manufacturer's protocol. The bound protein was separated by centrifugation (200 × g for 2 min at 4 °C) where the unbound-flowthrought was removed and the beads were washed extensively four times using lysis buffer and centrifugation at 200 ×g for 2 min at 4 °C. Finally, the beads were resuspended in 40 µl of sample buffer 2× and 20 µl were loaded on gel.

**Proteomics**. Immunoprecipitated proteins were suspended in 1 M urea, 50 mM ammonium bicarbonate, pH 7.8, and heated to 50 °C for 20 min. Proteins were reduced with 2 mM DTT, alkylated with 15 mM iodoacetamide, and digested overnight with trypsin. The resulting peptides were desalted on Waters Sep-Pak C18 cartridges. Peptides were measured by nano-LC-MS/MS on a Fusion Orbitrap (Thermo Fisher Scientific). Peptides were separated online by reverse phase chromatography using a heated 50 °C 30 cm C18 columns (75 mm ID packed with Magic C18 AQ 3 µM/100A beads) in a 280 min gradient (1–45% acetonitrile with 0.1% formic acid) separated at 250 nl min⁻¹. The fusion was operated in data-dependent mode with the following settings: 30,000 resolution, 350–1500 m/z full scan, top speed 2 s, and an 1.8 m/z isolation window. Identification and label free quantification of peptides were done with MaxQuant 1.5.7.4 using a 1% FDR against the human Swiss-Prot/TrEMB database downloaded from Uniprot on 2 June 2016. The databases contained forward and reverse human sequences as well as common contaminants. Peptides were searched using a match between run window of 2 min, 5 ppm mass error, a maximum of two missed cleavages, modification of cysteine residues (alkylation +57.0215 Da), methionine (oxidation +15.9949 Da), asparagine and glutamine (deamidation +0.9840), and acetylation of protein n-termini (+42.0106 Da). Proteins that were significantly regulated between conditions were identified using a permutation-based t-test (S1, FDR 5%) in Perseus 1.5.6. We analyzed two biological experiments with 3–4 technical replicates per condition.

**RNA extraction and RT-qPCR analysis**. RNA was extracted using Trizol (Life Technologies) according to the manufacturer's instructions. RNA samples were treated with Turbo DNase (ThermoFischer) and quantified using Nanodrop ND-1000 (Thermo Scientific). Reverse transcription was performed using Random Hexamers (Invitrogen) and Omniscript reverse transcription kit (Qiagen). Ten nanograms of cDNA were used to perform qRT-PCR using SYBR Green (Applied Biosystems) or TaqMan (Applied Biosystems). Primers used are listed in Supplementary Data 6D. Real-time RT-PCR analysis was performed on 7300 real-time PCR system (Applied Biosystems). ß-Actin was used as an endogenous control.

**RNA-seq and data analysis**. Total RNA for samples in this study were extracted with the Direct-zol RNA kit (Zymo Research) and libraries for Illumina sequencing were generated with the KAPA Stranded mRNA-Seq Kit (KAPA Biosystems). Sequencing was run on an Illumina HiSeq 2500 in the Fred Hutchinson Cancer Research Center Shared Resources Genomics facility. RNA-seq samples from this study and previously published studies[1,4,7,12,67,68] were aligned to hg19 using Tophat[69] (version 2.0.13). Gene-level read counts were quantified using featureCounts (24227677)[70] using Ensembl GRCh37 gene annotations. Raw sequencing data and read count per gene data can be accessed at the NCBI Gene Expression Omnibus. Batch effects were corrected using Combat[71]. Processed single-cell RNA-seq data from Nakamura et al.[33] were used. Only genes expressed above 10 Reads Per Million in three or more samples were kept. t-SNE[72] was performed with the Rtsne package, using genes with the top 20% variance across samples. Cluster labels from Nakamura et al.[33] were used.

**Small-molecule screen for naïve and primed hESC**. Compound sensitivity profiles of ELF1 naïve and ELF1 primed cells were compared in a high throughput screen of 160 approved and investigational oncology drugs described previously[73]. Composition of this customized drug panel includes 45 FDA-approved drugs while ~80% of the remaining compounds have been tested in early phase clinical trials. These drugs encompass a broad range of activities and target classes and include inhibitors of EGFR (n = 2), FLT3 (5), PI3K/AKT/mTOR (25), MEK/ERK (7), histone deacetylases (5), Bcl2 (3), NF-κB (2), Jak/Stats (2), PARPs (3), CDKs (7), HSP90 (2), CHKs (4), polo-like kinases (2), aurora kinases (2), hedgehogs (3), PKC (2), farnesyl transferases (2), GSK3 (2), proteosome components (1), additional kinases (29), retinoids (3), and drugs that act on other or unknown targets (12), and >80% have been tested in early phase clinical trials. Cells were seeded onto tissue culture-treated 384-well plates that were previously prepared by a Ghosting Protocol[74] where all cellular manipulations were performed on a BioTek EL406 liquid handler. Wells of the 384-well plate were coated with Matrigel (Corning) per the manufacturer's recommended procedure and kept at room temperature at least

15 min prior to the addition of MEF. For naïve cells, 386-well plates in triplicate were seeded with γ-irradiated MEF. The feeder layer was allowed to adhere overnight. The next day feeder extracellular matrix was generated through lysis[74]. Two thousand Elf1 naïve cells per well were seeded the same day onto the extracellular matrix in MEF conditioned hESC medium (supplemented with 2iL-I-F or FGF; naïve and primed, respectively) and 4000 cells per well Elf1 primed cells were also seeded in triplicate and both naïve and primed allowed to adhere overnight at 37 °C, 5% $CO_2$ in 50 μl volume. After 24 h incubation at 37 °C, 5% $CO_2$, media was replaced with fresh media. Immediately following the media exchange, compounds (50 nl) were added at concentrations ranging from 5 pM to 100 μM using the CyBio CyBi-Well Vario and incubated at 37 °C, 5% $CO_2$ for 72 h and viability was assessed using CellTiter Glo (Promega) with quantitation of luminescence derived from intracellular ATP. Resulting dose curves were fitted to a 4 Parameter Logistic Dose Response Model using idbs' XLFit. CellTiter-Glo (Promega) is dispensed into the individual wells with the WellMate and following 20 min incubation on an orbital shaker, luminescence is measured on a Perkin Elmer EnVision Multi-label plate reader. Percentage cell viability is reported as relative to the DMSO solvent control. After 4 days, the results were obtained. $IC_{50}$ values were calculated by fitting data using least-squares method to the standard four-parameter logistic model where: $Y = (Y_{min} + (Y_{max}/(1 + ((X/IC_{50})^{Slope}))$, and $Y = \%$ viability, $Y_{min} =$ minimal % viability, $Y_{max} =$ maximal % viability, $X =$ compound concentration, $IC_{50} =$ concentration of compound exhibiting 50% inhibition of cellular viability, Slope = the slope of the resultant curve. Curve fitting was performed using idbs XLFit software, an Addin for Microsoft Excel. Subsequent analysis was performed using Tibco's Spotfire software.

**Reproducibility of experiments**. The number of independent experiments for each figure panel is described in the corresponding figure legend and primary data are available in Supplementary Data 7. For all western blots presented in the manuscript, there were at least three independent experiments.

There is no estimate of variation in each group of data and the variance is similar between the groups. No statistical method was used to predetermine sample size. The experiments were not randomized. The investigators were not blinded to allocation during experiments and outcome assessment. RNA samples with 260 nm/280 nm < 1.80 were discarded.

## Data availability

RNA-seq and ChIP-seq data sets generated for this study are available from the NCBI GEO database under accession number GSE122118. The mass spectrometry proteomics data have been deposited to the ProteomeXchange Consortium via the PRIDE[75] partner repository with the dataset identifier PXD011736. A reporting summary is available as a Supplementary File. The source data underlying Figs. 1e–g, 2c, g, h, 3d–f, g, i–k, 4e, g, h, k, l, 5b–g and Supplementary Figs. 3B, 4D, 4J, 4L, 5B–C, 7A are provided as a Source Data file.

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

## Acknowledgements

We thank members of the Ruohola-Baker laboratory for helpful discussions throughout this work. We thank Jennifer Hesson, Ammar Alghadeer, and Asis Hussein for technical help, and Dr. Rudolf Jaenisch for providing WIBR3 5iLA cells. The small-molecule screen was done in Quellos High Throughput Screening Core, Institute for Stem Cell and Regenerative Medicine, University of Washington School of Medicine. This work is supported in part by the University of Washington's Proteomics Resource (UWPR95794). R.T.M. is an Investigator, and A.M.R. is Associate, of the HHMI. This work is supported by the ISCRM Innovation Pilot Award for J.M. and grants from the National Institutes of Health R01GM097372, R01GM97372-03S1, and R01GM083867 for H.R.-B., 1P01GM081619 for C.B.W. and H.R.-B., and the NHLBI Progenitor Cell Biology Consortium (U01HL099997; UO1HL099993) for P.J.P., C.B.W. and H.R.-B.

## Author contributions

J.M and H.R.-B. conceived and designed the experiments. J.M., D.D., D.K., C.C., S.S, L.S., A.F, T.B., A.M.A., S.L., F.A., S.B. performed the experiments and/or analyzed the resulting data. Y.W. performed the bioinformatic analysis of the ChIP-seq, RNA-seq, and CRISPR screen data. A.M.R. performed mass spectrometry analysis. D.H, R.T.M., C.B.W. and P.J.P. provided materials and advice. J.M. and H.R.B. wrote the manuscript with input from the other authors.

## Additional information

**Competing interests:** The authors declare no competing interests.

