## [Peer Review File · Nature Communications]

Reviewers' comments:

Reviewer #1 (Remarks to the Author):

In this manuscript Ruohola-Baker and co-workers explore the mechanism by which human pluripotent stem cells exit from naive pluripotency and enter the primed state. They perform a screen based on metabolic and epigenetic properties of primed vs naive hESCs to identify genes potentially required for the state transition. As expected, one of the genes they uncover is Folliculin, which has been shown to play an important role in early postimplantation development in vivo and exit from naive pluripotency in vitro in mice. Various targets of FLCN have been explored previously in mice, including TFE3. The authors perform some nice techniques to confirm the requirement for FLCN in the phosphorylation and other post transcriptional modifications of TFE3. They also search for TFE3 targets and identify several Wnt family members. They make use of Wnt inhibition to confirm its functional requirement for the exit from naive pluripotency. Interestingly, when performing molecular profiling of the FLCN KO cells, they find that most, but not all of the changes normally occurring during the naive to primed transition are represented. However, the transcriptome of primate naive pluripotent cells is largely conserved in the FLCN KO cells. As expected, they reveal a role for targets of the mTORC pathway in the transition from naive to primed pluripotency, via FLCN. The study is well conceived and executed on the whole. There are several points that could be added/clarified to enhance the message of the study:

1. In Line 91 the authors state that Oct4 is a marker specifically of naive pluripotency, which is incorrect.
2. Line 106: please define NNMT
3. The section between lines 113 and 120 is confusing and could be rewritten more clearly.
4. The legend to Fig. 1 does not include enough information for the uninitiated reader to understand the figure, so should be embellished.
5. Fig. 2D is meaningless in its current state. An example of primed cells should be included if the authors wish to communicate that this figure corroborates the naive status of FLCN KO cells. I suspect that the primed cells will also show nuclear Oct4 staining.
6. Lines 261-264 could be more clearly explained.
7. Fig. 4I needs to include the WT levels of IDO1 and DNMT3L.
8. There is no mention (as far as I can see) of the number of passages for the lines used in any of the experiments. It is important to include this information to give the reader an indication of how well the various states of pluripotency are maintained in the various conditions.

Reviewer #2 (Remarks to the Author):

This study examined factors important for the exit of pluripotency in hESCs.

The study confirmed previous observations made in mESCs that the FLCN-TFE3 axis plays a key role in the regulation of maintenance and/or exit of pluripotency. Similar to the observations in mESCs, loss of the functional FLCN complex causes TFE3 nuclear accumulation and its activation leading to the blockade of exit from native pluripotency. The results demonstrated that the activation of the Wnt signaling, a known target of TFE3, play an important role in maintaining the pluripotency in hESCs lacking the expression of FLCN. It has been previously demonstrated that nuclear TFE3 localization is negatively regulated by cellular mTORC1 activity. To determine the molecular mechanisms by which loss of FLCN causes a constitutive activation of TFE3, the authors tried to identify FLCN interacting proteins and found that FLCN may inhibit mTORC2 through its binding to Sin1, a specific component of mTORC2, in native hESCs. The authors proposed a model where while active TFE3-Wnt axis maintains a native state of pluripotency, FLCN may activate mTORC1 on the lysosomal membrane leading to mTORC1-dependent TFE3 inhibition, a process important for the exit of pluripotency in hESCs.

Although many aspects in this paper were previously published (roles of FLCN, TSC2, TFE3, and Wnt in the regulation of pluripotency of ES cells and a relationship of these factors for the exit of pluripotency) and predicted from the literatures (a regulation of TFE3 by mTORC1), the studies used powerful genetic screening assays and unbiased proteomics to identify the importance of the TFE3-Wnt axis for the maintenance native pluripotency in hESs, which is inhibited by FLCN for the exit of pluripotency. Weakness of this manuscript include ambiguous points related with the molecular mechanisms of FLCN-dependent TFE3 inhibition and mTORC2 inactivation. Generally, there are many rooms to improve the quality of biochemical data and more data will be required to propose the authors' model.

Specific comments:

1. In Fig. 4E, what is another post-translational modification of TFE3 occurred in FLCN KO cells? Is this modification is regulated by mTORC1 or/and mTORC2, and important for the regulation of FLCN's nuclear-cytosolic shuttling?
2. In Fig. 5B, it remains largely unclear whether cellular mTORC1 and mTORC2 activity are indeed enhanced in FLCN KO cells without determining levels of total S6 and Akt proteins. It is also important to determine levels of p4EBP1/4EBP1, pS6K1/S6K, and p308Akt/Akt. In addition, to evaluate cellular PI3K activity in FLCN KO cells is informative.
3. In Fig. Sup 3C, did the authors indeed examine the expression of ESRRB protein in the hESc and its role in the exit from pluripotency in FLCN KO cells?
4. In Fig. Sup 4C, the data is not convincing without showing the amount of IPed FLCN and Sin1 in lysates.
5. In Fig. 5C, please evaluate the effect of specific mTORC1 inhibition on TFE3 cellular localization to strengthen the model proposed in this paper.
6. In Fig. 5D, does authors have any supportive data showing that mTORC2 functions upstream of mTORC1 in the hESc? Previous genetic studies in MEFs clearly demonstrated that ablation of mTORC2 has little effect on cellular mTORC1 activity since suffice Akt activity is maintained in mTORC2 knockout MEFs.
7. In related to the comment 6, does the activity of mTOR2 have any impact on a ground-state pluripotency or the exit from pluripotency in normal and FLCN KO hESc?

Reviewer #3 (Remarks to the Author):

The manuscript of Mathieu and colleagues describes the identification of FLCN, through a genome wide CRISPR-Cas9 screen, as a regulator of human naïve pluripotency. FLCN was previously identified as a critical regulator of exit from naïve pluripotency in mouse ES cells, acting by controlling the nuclear localization of Tfe3, which in turns activates the transcription factor Esrrb. Here the authors showed that the same regulatory axis controls exit from human naïve pluripotency, but proposed the Wnt signaling pathway as the downstream effector of Tfe3.

The experiments are generally well performed, yet the presentation of the experiments is sometimes a bit confusing.

The conclusions are interesting, and the other regulators identified by the screen could be of interest for the community.

In order to improve the impact of this work I advise the authors to follow these points:

- 1) the authors identified TFCP2L1/LBP9, KLF4 and DNMT3L as a downstream target of FLCN-TFE3 in multiple experiments (Figure 2 and 3), and provided very interesting results, such as the Transposable elements profiling in Figure S2F. The genes KLF4 and TFCP2L1 have been shown to be critical for

maintenance of human naive pluripotency by Takashima and colleagues, therefore they represent obvious candidates to test as effectors downstream of TFE3. It is therefore a bit surprising that the authors shifted their focus to the Wnt pathway and performed IWP2 treatment experiments.

Are KLF4 and TFCEP2L1 critical for the differentiation defects observed in FLCN KO cells? Is their downregulation by shRNA sufficient to rescue the effects of FLCN knockout?

2) the implication of the Wnt pathway is supported by the induction of some Wnt pathway components in FLCN KO cells. Notably, 3 of the 4 ligands are non-canonical Wnt pathway components. The authors stressed multiple times the importance of Wnt as a pathway regulated by Folliculin (e.g. in the title, abstract, discussion), but have not formally proved that Wnt is negatively regulated by Folliculin.

Are the canonical Wnt pathway target genes AXIN2 and TROY upregulated in differentiating FLCN KO cells? Is the BAR-Venus reporter used in Xu et al, PNAS 2016 more active in differentiating FLCN KO? Are beta-catenin levels or localization affected by FLCN?

3) Esrrb was previously described as a Tfe3 target in mouse ES cells. It was then reported by multiple groups that ESRRB is not expressed in human naive cells, yet its exogenous expression could sustain pluripotency (reviewed by Festuccia, Owens and Navarro, FEBS Letters 2017). The authors presented some results that are somehow inconsistent: in Figure 4F cells in 5iLA and in 2iL-I-F display very low ESRRB expression; in contrast Figure S3C shows that 5iLA cells express ESRRB >150 times more than 2iL-I-F.

Moreover, FLCN KO cells in 2iL-I-F show an upregulation of >50 fold of ESRRB, which is rescued by FLCN expression.

Such robust upregulations of ESRRB could indicate a functional role, why was this not tested?

Moreover, why are different datasets so inconsistent? Is it a normalization artifact (i.e. 150 times more than nothing could still be nothing)?

Minor point:

-sometimes the presentation of results is a bit confusing, for example Figure 3J shows that 3 naive markers are upregulated in FLCN KO during exit. It is not clear why the expression of such markers should be higher at day 7 compared to day 4. What is exactly the "WT FGF" sample?

The authors should show the initial levels of expression in 5iLA cells, and how such markers are downregulated over time in WT and FLCN KO cells.

-In general figure legends should provide more details about how data were normalized and presented (see also point 3).

-The LBP9 nomenclature is confusing, TFCEP2L1 is the official gene name widely used since its discovery as a naive pluripotency regulator.

Referees' comments:

Reviewer #1 (Remarks to the Author):

In this manuscript Ruohola-Baker and co-workers explore the mechanism by which human pluripotent stem cells exit from naive pluripotency and enter the primed state. They perform a screen based on metabolic and epigenetic properties of primed vs naive hESCs to identify genes potentially required for the state transition. As expected, one of the genes they uncover is Folliculin, which has been shown to play an important role in early postimplantation development in vivo and exit from naive pluripotency in vitro in mice. Various targets of FLCN have been explored previously in mice, including TFE3. The authors perform some nice techniques to confirm the requirement for FLCN in the phosphorylation and other post transcriptional modifications of TFE3. They also search for TFE3 targets and identify several Wnt family members. They make use of Wnt inhibition to confirm its functional requirement for the exit from naive pluripotency. Interestingly, when performing molecular profiling of the FLCN

KO cells, they find that most, but not all of the changes normally occurring during the naive to primed transition are represented. However, the transcriptome of primate naive pluripotent cells is largely conserved in the FLCN KO cells. As expected, they reveal a role for targets of the mTORC pathway in the transition from naive to primed pluripotency, via FLCN. The study is well conceived and executed on the whole. There are several points that could be added/clarified to enhance the message of the study:

1. In Line 91 the authors state that Oct4 is a marker specifically of naive pluripotency, which is incorrect.

Response: We thank the reviewer for pointing this out. We apologize for the mistake and have now corrected it.

2. Line 106: please define NNMT

Response: NNMT has now been defined in the text.

3. The section between lines 113 and 120 is confusing and could be rewritten more clearly.

Response: We have now rewritten this section more clearly.

4. The legend to Fig. 1 does not include enough information for the uninitiated reader to understand the figure, so should be embellished.

Response: We have modified the legend of Fig.1 to include more details.

5. Fig. 2D is meaningless in its current state. An example of primed cells should be included if the authors wish to communicate that this figure corroborates the naive status of FLCN KO cells. I suspect that the primed cells will also show nuclear Oct4 staining.

Response: This figure shows that in absence of FLCN, naïve hESC do not lose the pluripotency marker Oct4. This point is now more clearly explained in the text.

6. Lines 261-264 could be more clearly explained.

Response: This paragraph has been rewritten more clearly.

7. Fig. 4I needs to include the WT levels of IDO1 and DNMT3L.

Response: We have now performed RNAseq analysis of FLCN KO cells exiting naïve pluripotency and treated with either DMSO or the Wnt inhibitor IWP2. Principal component analysis reveals that inhibition of Wnt rescues the FLCN KO phenotype (Fig.4M). We now show the expression of several naïve and primed markers (SupplFig.5A-C), including IDO1 and DNMT3L, in wild type and experimental conditions, as suggested by the reviewer.

8. There is no mention (as far as I can see) of the number of passages for the lines used in any of the experiments. It is important to include this information to give the reader an indication of how well the various states of pluripotency are maintained in the various conditions.

Response: We agree with the reviewer and have now added this information.

Reviewer #2 (Remarks to the Author):

This study examined factors important for the exit of pluripotency in hESCs. The study confirmed previous observations made in mESCs that the FLCN-TFE3 axis plays a key role in the regulation of maintenance and/or exit of pluripotency. Similar to the observations in mESCs, loss of the functional FLCN complex causes TFE3 nuclear accumulation and its activation leading to the blockade of exit from native pluripotency. The results demonstrated that

the activation of the Wnt signaling, a known target of TFE3, play an important role in maintaining the pluripotency in hESCs lacking the expression of FLCN. It has been previously demonstrated that nuclear TFE3 localization is negatively regulated by cellular mTORC1 activity. To determine the molecular mechanisms by which loss of FLCN causes a constitutive activation of TFE3, the authors tried to identify FLCN interacting proteins and found that FLCN may inhibit mTORC2 through its binding to Sin1, a specific component of mTORC2, in native hESCs. The authors proposed a model where while active TFE3-Wnt axis maintains a native state of pluripotency, FLCN may activate mTORC1 on the lysosomal membrane leading to mTORC1-dependent TFE3 inhibition, a process important for the exit of pluripotency in hESCs.

Although many aspects in this paper were previously published (roles of FLCN, TSC2, TFE3, and Wnt in the regulation of pluripotency of ES cells and a relationship of these factors for the exit of pluripotency) and predicted from the literatures (a regulation of TFE3 by mTORC1), the studies used powerful genetic screening assays and unbiased proteomics to identify the importance of the TFE3-Wnt axis for the maintenance native pluripotency in hESCs, which is inhibited by FLCN for the exit of pluripotency. Weakness of this manuscript include ambiguous points related with the molecular mechanisms of FLCN-dependent TFE3 inhibition and mTORC2 inactivation. Generally, there are many rooms to improve the quality of biochemical data and more data will be required to propose the authors' model.

Specific comments:

1. In Fig. 4E, what is another post-translational modification of TFE3 occurred in FLCN KO cells? Is this modification is regulated by mTORC1 or/and mTORC2, and important for the regulation of FLCN's nuclear-cytosolic shuttling?

Response: We agree with the reviewer that post-translational modification of TFE3 occurring in FLCN KO cells is very interesting. While we are in the process of further analyzing this modification, today we still do not know the cause for this MW shift. We have however, tested if this modification is regulated by mTORC1 or/and mTORC2, as suggested by the reviewer. No obvious, comparable MW shift was observed in these conditions.

2. In Fig. 5B, it remains largely unclear whether cellular mTORC1 and mTORC2 activity are indeed enhanced in FLCN KO cells without determining levels of total S6 and Akt proteins. It is also important to determine levels of p4EBP1/4EBP1, pS6K1/S6K, and p308Akt/Akt. In addition, to evaluate cellular PI3K activity in FLCN KO cells is informative.

Response: We agree with the reviewers #2 and 3. As described below, we have now shown that specific inhibition of mTORC1 by Rapamycin (evidenced by decreased of pS6 but not pAkt) represses TFE3 exit from nucleus in naïve-to-primed transition assay (Fig.5C-D), suggesting that specific mTORC1 inhibition affects TFE3 cellular localization as proposed in the model. TORC2 specific mutant, SIN1 does not affect TFE3 localization. We also show using the inhibitors that while TORC1 specific inhibitor affects P-S6, a stronger inhibition is achieved using TORC1/TORC2 inhibitor, INK-128. These data suggest that mTORC2 functions upstream of mTORC1 in the hESC, as proposed in the model. Total levels of S6 and Akt proteins are now determined in these analyses.

3. In Fig. Sup 3C, did the authors indeed examine the expression of ESRRB protein in the hESC and its role in the exit from pluripotency in FLCN KO cells?

Response: We have now analyzed the expression of ESRRB by Western blot and, as expected from the QPCR analysis, showed that ESRRB can be detected in hESC (Fig.4G). However,

importantly, ESRRB is not significantly upregulated in Flcn mutant lines during naïve-to-primed transition (SupplFig.4C; Fig.4G, 4I). Therefore, the lack of exit in Flcn null lines cannot be caused by TFE3 dependent expression of ESRRB. To further analyze ESRRB function in hESC, we generated an ESRRB mutant line using the CRISPR Cas9 system (suppl fig4D-G) and showed that ESRRB mutant cells still express pluripotency markers Oct4 and Nanog as well as the naïve marker DNMT3L (suppl fig4F-G). In addition, ESRRB mutant did not rescue the FLCN KO phenotype during the exit from the naïve pluripotent state (Fig.4H).

4. In Fig. Sup 4C, the data is not convincing without showing the amount of IPed FLCN and Sin1 in lysates.

Response: We thank the reviewer for pointing this out. We have now added to the figure the amount of FLCN and SIN1 IPed (PD) and in the lysates (T) (SupplFig.6A).

5. In Fig. 5C, please evaluate the effect of specific mTORC1 inhibition on TFE3 cellular localization to strengthen the model proposed in this paper.

Response: We have now performed new experiments and shown that specific inhibition of mTORC1 by Rapamycin (evidenced by decreased of pS6 but not pAkt) represses TFE3 exit from nucleus in naïve-to-primed transition assay (Fig.5C-D), suggesting that specific mTORC1 inhibition affects TFE3 cellular localization as proposed in the model.

6. In Fig. 5D, do authors have any supportive data showing that mTORC2 functions upstream of mTORC1 in the hESC? Previous genetic studies in MEFs clearly demonstrated that ablation of mTORC2 has little effect on cellular mTORC1 activity since suffice Akt activity is maintained in mTORC2 knockout MEFs.

Response: As described above, we have now shown that specific inhibition of mTORC1 by Rapamycin (evidenced by decreased of pS6 but not pAkt) represses TFE3 exit from nucleus in naïve-to-primed transition assay (Fig.5C-D), suggesting that specific mTORC1 inhibition affects TFE3 cellular localization as proposed in the model. We also show using these inhibitors that while TORC1 specific inhibitor affects P-S6, a stronger inhibition is achieved using TORC1/TORC2 inhibitor, INK-128. These data suggest that mTORC2 functions upstream of mTORC1 in the hESC, as proposed in the model.

7. In related to the comment 6, does the activity of mTOR2 have any impact on a ground-state pluripotency or the exit from pluripotency in normal and FLCN KO hESC?

Response: To specifically inhibit TORC2, we tried to generate KO of SIN1, an essential component of mTORC2 using CRISPR-Cas9 genome editing. Despite the evidence of a highly mutated pool and the screening of more than 40 clones, we were not able to obtain a SIN1 null line, suggesting that SIN1 may be essential for the survival of hESC (suppl. Fig6B-E). We however generated lines exhibiting reduced levels of SIN1. Reduction of SIN1 does not eliminate the pluripotency marker Oct4 in naïve hESC (Suppl.Fig.6C,E) or during the exit of naïve state (Fig4C).

Reviewer #3 (Remarks to the Author):

The manuscript of Mathieu and colleagues describes the identification of FLCN, through a genome wide CRISPR-Cas9 screen, as a regulator of human naïve pluripotency. FLCN was previously identified as a critical regulator of exit from naïve pluripotency in mouse ES cells, acting by controlling the nuclear localization of Tfe3, which in turns activates the transcription

factor Esrrb. Here the authors showed that the same regulatory axis controls exit from human naïve pluripotency, but proposed the Wnt signaling pathway as the downstream effector of Tfe3. The experiments are generally well performed, yet the presentation of the experiments is sometimes a bit confusing.

The conclusions are interesting, and the other regulators identified by the screen could be of interest for the community.

In order to improve the impact of this work I advise the authors to follow these points:

1) the authors identified TFCE2L1/LBP9, KLF4 and DNMT3L as a downstream target of FLCN-TFE3 in multiple experiments (Figure 2 and 3), and provided very interesting results, such as the Transposable elements profiling in Figure S2F. The genes KLF4 and TFCE2L1 have been shown to be critical for maintenance of human naïve pluripotency by Takashima and colleagues, therefore they represent obvious candidates to test as effectors downstream of TFE3. It is therefore a bit surprising that the authors shifted their focus to the Wnt pathway and performed IWP2 treatment experiments.

Are KLF4 and TFCE2L1 critical for the differentiation defects observed in FLCN KO cells? Is their downregulation by shRNA sufficient to rescue the effects of FLCN knockout?

Response: We agree with the reviewer that KLF4 and TFCE2L1 are critical for naïve human pluripotency and that it is therefore important to study their mechanisms of regulation. However, while both KLF4 and TFCE2L1 are up-regulated in FLCN KO in the naïve state, they are not upregulated during the exit of 2iL-I-F naïve pluripotency (KLF4: 664rpkm in WT 7D TeSR, 554rpkm in FLCN KO 7D TESR- 0.83 fold; TFCE2L1: 633rpkm in WT 7D TeSR, 672rpkm in FLCN KO 7D TESR- 1.06 fold). This suggests that TFE3 does not regulate KLF4 and TFCE2L1 in FLCN KO cells during the exit of the 2iL-I-F naïve pluripotent state, and therefore these genes cannot be causal for Flcn KO phenotypes that are the focus of this paper.

2) the implication of the Wnt pathway is supported by the induction of some Wnt pathway components in FLCN KO cells. Notably, 3 of the 4 ligands are non-canonical Wnt pathway components. The authors stressed multiple times the importance of Wnt as a pathway regulated by Folliculin (e.g. in the title, abstract, discussion), but have not formally proved that Wnt is negatively regulated by Folliculin.

Are the canonical Wnt pathway target genes AXIN2 and TROY upregulated in differentiating FLCN KO cells? Is the BAR-Venus reporter used in Xu et al, PNAS 2016 more active in differentiating FLCN KO? Are beta-catenin levels or localization affected by FLCN?

Response: We have now shown that during the exit of the naïve pluripotent state, FLCN KO not only up-regulates the expression of Wnt ligands but also upregulates the expression of Wnt pathway targets, including the canonical target TROY (RNAseq, qPCR, Western; Fig.4I-L). We also now show that the expression of beta-catenin protein is up-regulated in FLCN KO 7D TeSR compared to WT 7D TeSR (Fig.4L). In addition, we have now performed RNAseq analysis of FLCN KO cells exiting naïve pluripotency with DMSO or with the Wnt inhibitor IWP2. Principal component Analysis using these data reveals that inhibition of Wnt rescues the FLCN KO phenotype (Fig.4M).

We agree with the reviewer that the BAR-Venus reporter that we have used in the past in collaboration with Dr Moon's laboratory is a good tool to monitor Wnt activity in hESC. However, unfortunately this reporter assay cannot be used in our early exit-assay due to the long half life of the fluorescence protein. FLCN KO prevent the naïve cells to transition to the primed stage and cannot be kept in primed TeSR conditions for several passages (that would be necessary to observe a decreased Wnt activity with the BAR reporter in primed hESC). It is therefore difficult to assess the difference between WT and FLCN KO in Wnt activity using the BAR reporter at early time points. However, we feel the new analysis presented in Fig.4F-M strongly argue Wnt-

pathway involvement as a Folliculin target in hESC.

3) *Esrrb* was previously described as a Tfe3 target in mouse ES cells. It was then reported by multiple groups that *ESRRB* is not expressed in human naïve cells, yet its exogenous expression could sustain pluripotency (reviewed by Festuccia, Owens and Navarro, FEBS Letters 2017). The authors presented some results that are somehow inconsistent: in Figure 4F cells in 5iLA and in 2iL-I-F display very low *ESRRB* expression; in contrast Figure S3C shows that 5iLA cells express *ESRRB* >150 times more than 2iL-I-F. Moreover, *FLCN* KO cells in 2iL-I-F show an upregulation of >50 fold of *ESRRB*, which is rescued by *FLCN* expression.

Such robust upregulations of *ESRRB* could indicate a functional role, why was this not tested? Moreover, why are different datasets so inconsistent? Is it a normalization artifact (i.e. 150 times more than nothing could still be nothing)?

Response: As described above, we have now analyzed the expression of *ESRRB* by Western blot and, as expected from the QPCR analysis, showed that *ESRRB* can be detected in hESC (Fig.4G). However, importantly, *ESRRB* is not significantly upregulated in *Flcn* mutant lines during naïve-to-primed transition (SupplFig.4C; Fig.4G, 4I). Therefore, the lack of exit in *Flcn* null lines cannot be caused by TFE3 dependent expression of *ESRRB*. To further analyze *ESRRB* function in hESC, we generated an *ESRRB* mutant line using the CRISPR Cas9 system (suppl fig4D-G) and showed that *ESRRB* mutant cells still express pluripotency markers Oct4 and Nanog as well as the naïve marker DNMT3L (suppl fig4F-G). In addition, *ESRRB* mutant did not rescue the *FLCN* KO phenotype during the exit from the naïve pluripotent state (Fig.4H).

Minor point:

-sometimes the presentation of results is a bit confusing, for example Figure 3J shows that 3 naïve markers are upregulated in *FLCN* KO during exit. It is not clear why the expression of such markers should be higher at day 7 compared to day 4. What is exactly the “WT FGF” sample?

The authors should show the initial levels of expression in 5iLA cells, and how such markers are downregulated over time in WT and *FLCN* KO cells.

Response: We have now clarified this presentation.

-In general figure legends should provide more details about how data were normalized and presented (see also point 3).

Response: We have now modified the figure legends to provide more details.

-The LBP9 nomenclature is confusing, *TFCP2L1* is the official gene name widely used since its discovery as a naïve pluripotency regulator.

Response: We have used *TFCP2L1* throughout the paper.

Reviewers' comments:

Reviewer #2 (Remarks to the Author):

In the revised Sup 6A, it is difficult to evaluate the specific FLCN-mSin1 interaction in naive hESC cells because the amount of IPed FLCN from the native hESC cells seem to be higher than that from TsSR cells. Please show much lighter exposure of the firms.

In Fig. 5D and Sup6F, it is well known that rapamycin cannot inhibit the phosphorylation of mTORC1 substrates completely as it is not a direct kinase inhibitor and the effect of rapamycin on 4EBP1 (small substrate) phosphorylation is moderate. In contrast, ATP-competitive mTOR kinase inhibitors including INK completely inhibit 4EBP1 phosphorylation. Thus, the stronger inhibitory effect of INK128 on the phosphorylation of mTORC1 does not strongly support the idea that mTORC2 acts upstream of mTORC1.

This reviewer feels that the authors need to demonstrate more stronger data that FLCN indeed interacts with mTORC2 (not only mSin1) and inhibits its function in the naive cells to strengthen the model proposed. It is possible that FLCN interacts with mSin1, which is not a part of mTORC2, in the naive cells.

Reviewer #3 (Remarks to the Author):

Remarks to the Author:

The authors addressed most of the points raised by reviewers, however several points still require clarification:

1-The Authors first claimed that KLF4 and TFCP2L1 are regulated by FLCN in multiple experiments, but they have not tested their functional requirement, because there is no difference in their expression during differentiation (data not shown, but present in the RNAseq). However, they do show that those two markers are unregulated during differentiation, in figure 3K and S3B.

Why did the authors give more importance to one dataset (not shown in the paper) over multiple independent datasets and conclude that KLF4 and TFCP2L1 are not important?

2-The transition from naive to primed is assessed with different markers in different experiments. The authors should use the same markers in all experiments. The use of markers that have been shown to be expressed in human pre-implantation blastocysts and are functionally important (e.g. KLF4, TFCP2L1, KLF17) would be preferable.

3- ESRRB has been "mutated" by CAS9, yet the protein is still present (Figure 4H) and the authors concluded ESRRB is not relevant. Such conclusion is based on the expression of a single marker (HIF1a, see point 2).

It is not clear from the text whether such cells are double KO cells, which is unlikely given that the Western Blot in Figure 4H shows a clear ESRRB band, or just hypomorphic alleles.

Also, from Figure S4G it is not clear which clone has been used for the experiment presented in Figure 4H.

All these points should be clarified.

4-The claim that ESRRB "reduction during naïve-to-primed hESC transition does not rescue the Flcn

KO phenotypes" is based on Figure 4H, where only two markers unregulated in primed cells are not affected by ESRRB reduction.

This reviewer suggests to rephrase such claim as "ESRRB reduction does not affect the expression of the primed markers HIF1a and LDHA".

Response to Reviewers' comments:

We have now performed new experiments in light of the reviewers' constructive comments.

We thank the reviewers for their insightful comments and feel that the revised manuscript is now substantially improved. In light of these new findings, we feel that the paper is now suitable for publication in Nature Communications. We hope you agree.

Thank you for considering our manuscript.

Point-to-point response to reviewers' comments follows.

Sincerely,
Hannele

Reviewers' comments:

Reviewer #2 (Remarks to the Author):

In the revised Sup 6A, it is difficult to evaluate the specific FLCN-mSin1 interaction in naive hESC cells because the amount of IPed FLCN from the native hESC cells seem to be higher than that from TsSR cells. Please show much lighter exposure of the films.

Response: As suggested by the reviewer, we now show a much lighter exposure of the film (Suppl.Fig.6A). The light exposure shows that the amount of IPed FLCN is the same in naive hESC and TeSR treated cells.

In Fig. 5D and Sup6F, it is well known that rapamycin cannot inhibit the phosphorylation of mTORC1 substrates completely as it is not a direct kinase inhibitor and the effect of rapamycin on 4EBP1 (small substrate) phosphorylation is moderate. In contrast, ATP-competitive mTOR kinase inhibitors including INK completely inhibit 4EBP1 phosphorylation. Thus, the stronger inhibitory effect of INK128 on the phosphorylation of mTORC1 does not strongly support the idea that mTORC2 acts upstream of mTORC1.

Response: We now show that mutation of SIN1, an essential component of mTORC2 not only results in reduced phosphorylation of mTORC2 target, p(S473)Akt, but also reduced mTORC1 target PS6, suggesting that mTORC2 functions upstream of a branch of mTORC1 in primed hESC (Fig.5D-F). We agree that the drugs may have different level of inhibitor activity. However, we have now shown that in a different stage of pluripotency INK128 and Rapamycin give the same level of effect on mTOR pathway (in 5iLA), suggesting that the stronger inhibitor effect in primed stage in Fig.5C of INK128 might be due to a biological difference between these two stages. In summary, the genetic- and small molecule-based experiments in this manuscript suggest that mTORC2 acts upstream of a branch of mTORC1 during the exit of naïve pluripotency (TeSR).

This reviewer feels that the authors need to demonstrate more stronger data that FLCN indeed interacts with mTORC2 (not only mSin1) and inhibits its function in the naive cells to strengthen the model proposed. It is possible that FLCN interacts with mSin1, which is not a part of

mTORC2, in the naïve cells.

Response: We show that P-AKT(Ser473) is upregulated in FLCN mutant in naïve 5iLA and 2iL-I-F but not during the exit from naïve pluripotency (Fig.5B, Suppl.Fig.7A-B), indicating that FLCN represses mTORC2 activity in naïve hESC. In addition, to explore this question further, we have now also inhibited mTORC2 using INK128 and compared its effect to mTORC1 inhibitor Rapamycin in naïve hESC 5iLA (Fig.5C).

Reviewer #3 (Remarks to the Author):

Remarks to the Author:

The authors addressed most of the points raised by reviewers, however several points still require clarification:

1-The Authors first claimed that KLF4 and TFCEP2L1 are regulated by FLCN in multiple experiments, but they have not tested their functional requirement, because there is no difference in their expression during differentiation (data not shown, but present in the RNAseq). However, they do show that those two markers are unregulated during differentiation, in figure 3K and S3B.

Why did the authors give more importance to one dataset (not shown in the paper) over multiple independent datasets and conclude that KLF4 and TFCEP2L1 are not important?

Response: We have now tested the functional requirement of KLF4 and TFCEP2L1 in FLCN KO hESC using shRNA against the two genes. We found that KLF4 KD does partially rescue FLCN phenotype. However, since KLF4 is not a TFE3 target, its expression cannot explain the mechanism of FLCN function during the exit from naïve pluripotency. TFCEP2L1 KD slightly decreases the naïve marker KLF4 expression in FLCN KO cells exiting the naïve 5iLA state, but does not affect DNMT3L expression, suggesting that the reviewer is correct, TFCEP2L1 plays a minor role on FLCN phenotype. We now show the KD results (Suppl. Fig.4I-L) and discuss them in the manuscript.

2-The transition from naïve to primed is assessed with different markers in different experiments. The authors should use the same markers in all experiments. The use of markers that have been shown to be expressed in human pre-implantation blastocysts and are functionally important (e.g. KLF4, TFCEP2L1, KLF17) would be preferable.

Response: We have now used the naïve markers DNMT3L and/or KLF4 and TFCEP2L1 in all the naïve to primed transitions (Fig. 3, Fig. 4, Suppl Fig. 3, Suppl Fig. 4).

3- ESRRB has been "mutated" by CAS9, yet the protein is still present (Figure 4H) and the authors concluded ESRRB is not relevant. Such conclusion is based on the expression of a single marker (HIF1a, see point 2).

It is not clear from the text whether such cells are double KO cells, which is unlikely given that the Western Blot in Figure 4H shows a clear ESRRB band, or just hypomorphic alleles.

Also, from Figure S4G it is not clear which clone has been used for the experiment presented in Figure 4H.

All these points should be clarified.

Response: We have now generated new ESRRB KO lines in WIBR3 5iLA FLCN KO background and found that the KO of ESRRB during naïve-to-primed hESC transition does not

rescue the FLCN KO phenotypes based on the analysis of naïve hESC markers KLF4, TFCP2L1 and DNMT3L (Fig.4G-H, Suppl.Fig.4G-H). We also clarified in the text that the previously generated ESRRB mutant line is not a null and indicated which clone was used for the experiment presented in Suppl. Fig.4F (previously Fig.4H).

4-The claim that ESRRB "reduction during naïve-to-primed hESC transition does not rescue the Flcn KO phenotypes" is based on Figure 4H, where only two markers unregulated in primed cells are not affected by ESRRB reduction.

This reviewer suggests to rephrase such claim as "ESRRB reduction does not affect the expression of the primed markers HIF1a and LDHA".

Response: As suggested by the reviewer, we have now rephrased this section. We have also included our new data showing that ESRRB KO in naïve 5iLA FLCN KO does not decrease the expression of naïve markers DNMT3L, KLF4 and TFCP2L1 during the naïve to primed transition. Therefore, in human naïve ESC ESRRB KO does not rescue FLCN KO phenotype.

Reviewers' comments:

Reviewer #2 (Remarks to the Author):

The proposed model that FLCN inhibits mTORC2 through its direct interaction with mTORC2 in naïve hESC cells are still weak and more biochemical analyses will be required. The authors did not convincingly demonstrate specific interaction with FLCN and mTORC2. They should perform other immunoblots such as Rictor and mTOR and include appropriate controls to strengthen the proposed model that FLCN interacts with mTORC2 (not only Sin1). If FLCN binds to only Sin1 but not other components of mTORC2, then the authors need to investigate how the FLCN in naïve hESC cells downregulates mTORC2 activity.

In addition, as previously requested, the authors should present total S6, Akt, mTOR and pT308 blots in the Fig. 5B.

Since a previous study demonstrated FLCN inhibits both mTORC2 and mTORC1 likely through inhibiting PI3K, the proposed model in this study that FLCN directly interacts with mTORC2 to inhibit its function in naïve hESC must be fully delineated with convincing data sets.

Reviewer #3 (Remarks to the Author):

The Authors addressed all remaining points in a satisfactory way, thus I strongly support publication of the revised Manuscript.

Reviewers' comments:

Reviewer #2 :

The proposed model that FLCN inhibits mTORC2 through its direct interaction with mTORC2 in naïve hESC cells are still weak and more biochemical analyses will be required. The authors did not convincingly demonstrate specific interaction with FLCN and mTORC2. They should perform other immunoblots such as Rictor and mTOR and include appropriate controls to

strengthen the proposed model that FLCN interacts with mTORC2 (not only Sin1). If FLCN binds to only Sin1 but no other components of mTORC2, then the authors need to investigate how the FLCN in naïve hESC cells downregulates mTORC2 activity.

In addition, as previously requested, the authors should present total S6, Akt, mTOR and pT308 blots in the Fig. 5B.

Since a previous study demonstrated FLCN inhibits both mTORC2 and mTORC1 likely through inhibiting PI3K, the proposed model in this study that FLCN directly interacts with mTORC2 to inhibit its function in naïve hESC must be fully delineated with convincing data sets.

Response: In order to preserve the integrity of the mTORC complexes we modified the immuno-precipitation protocol as done before (0.3% CHAPS as the detergent; Kim et al., Cell 2002). While we were able to still detect a binding of FLCN to SIN1 in 2iLIF hESC using this modified protocol, we did not reliably detect other members of the mTORC2 complex. To further probe mTORC2 function in the process, we generated mutations in an mTORC2 specific component, RICTOR (Fig.5G). Importantly, RICTOR mutants show the same phenotypes as SIN1 mutants; reduction of P-AKT(S473) and P-S6 in the exit assay (Fig.5G). Furthermore, both SIN1 and RICTOR mutants show normal nuclear export of TFE3 during the exit from naïve state (Fig.5H). While SIN1 may also have other, mTOR2 independent functions, these data support the model in which FLCN inhibits mTORC2 through SIN1 in naïve hESC. Furthermore, while FLCN may have PI3K dependent functions in other cell types, we did not detect any significant changes on pAkt(T308) levels in FLCN KO compared to wild type hESC (Suppl.Fig.6C). These data suggest that in hESC FLCN directly inhibits mTORC2 activation rather than acting through PI3K to inhibit mTORC2.

We have now included immunoblots showing the total mTor, S6 and Akt in four different culture conditions of hESC: 5iLA, 2iL-I-F, FGF and TeSR (Fig5B, Suppl.Fig.6B), as well as pAkt(T308) in naïve and primed hESC states (Suppl.Fig.6C), as requested by the reviewer.

Reviewer #3:

The Authors addressed all remaining points in a satisfactory way, thus I strongly support publication of the revised Manuscript.

Response:

We thank the reviewer for her/his positive support

REVIEWERS' COMMENTS:

Reviewer #2 (Remarks to the Author):

The revised results suggest that FLCN may not directly interact with mTORC2, or FLCN may sequester Sin1 from the mTORC2. The genetic approach (Rictor KD) again places FLCN at upstream of mTORC2. These results should be more reflected in the model in Fig.5I.

The revised paper has been strengthened and would be appropriate for the journal.

A point-by-point response to the referee questions follows:

Reviewer #2 (Remarks to the Author):

The revised results suggest that FLCN may not directly interact with mTORC2, or FLCN may sequester Sin1 from the mTORC2. The genetic approach (Rictor KD) again places FLCN at upstream of mTORC2. These results should be more reflected in the model in Fig.5I.

The revised paper has been strengthened and would be appropriate for the journal.

Response: We have now stated in the text that FLCN may not directly interact with mTORC2 but rather may sequester Sin1 from this complex. We have now also reflected these results in the model.